# Drivers and sites of diversity in the DNA adenine methylomes of 93 *Mycobacterium tuberculosis* complex clinical isolates

Samuel J Modlin[1], Derek Conkle-Gutierrez[1], Calvin Kim[1], Scott N Mitchell[1], Christopher Morrissey[1], Brian C Weinrick[2], William R Jacobs[3], Sarah M Ramirez-Busby[1], Sven E Hoffner[1,4], Faramarz Valafar[1]*

[1]Laboratory for Pathogenesis of Clinical Drug Resistance and Persistence, San Diego State University, San Diego, United States; [2]Trudeau Institute, Saranac Lake, United States; [3]Department of Microbiology and Immunology, Albert Einstein College of Medicine, Bronx, United States; [4]Department of Public Health Sciences, Karolinska Institute, Stockholm, Sweden

**Abstract** This study assembles DNA adenine methylomes for 93 *Mycobacterium tuberculosis* complex (MTBC) isolates from seven lineages paired with fully-annotated, finished, de novo assembled genomes. Integrative analysis yielded four key results. First, methyltransferase allele-methylome mapping corrected methyltransferase variant effects previously obscured by reference-based variant calling. Second, heterogeneity analysis of partially active methyltransferase alleles revealed that intracellular stochastic methylation generates a mosaic of methylomes within isogenic cultures, which we formalize as 'intercellular mosaic methylation' (IMM). Mutation-driven IMM was nearly ubiquitous in the globally prominent Beijing sublineage. Third, promoter methylation is widespread and associated with differential expression in the Δ*hsdM* transcriptome, suggesting promoter HsdM-methylation directly influences transcription. Finally, comparative and functional analyses identified 351 sites hypervariable across isolates and numerous putative regulatory interactions. This multi-omic integration revealed features of methylomic variability in clinical isolates and provides a rational basis for hypothesizing the functions of DNA adenine methylation in MTBC physiology and adaptive evolution.

*For correspondence:
faramarz@sdsu.edu

Competing interests: The authors declare that no competing interests exist.

## Introduction

In 2017, tuberculosis (TB) killed 1.6 million people globally, the most of any infectious disease, despite significant TB control efforts and the availability of effective TB drugs (*WHO, 2017*). Multi-drug-resistant tuberculosis (MDR-TB) threatens control efforts and debilitates patients through a grueling and often ineffective treatment regimen (52% success) (*WHO, 2017*). The primary causative agent of TB, *M. tuberculosis*, has a low mutation rate, and is reported to evolve chiefly through single nucleotide polymorphisms (SNPs) (*Cohen et al., 2015*). However, subpopulations of the pathogen consistently persist through chemotherapeutics, eventually developing full antibiotic resistance (*Jain et al., 2016*). It is unclear how such a genetically static organism adapts so rapidly to drug treatment and varied immune pressures.

DNA methylation is a plausible yet scarcely explored alternative mechanism for phenotypic variation in *M. tuberculosis. M. tuberculosis* encodes three known DNA methyltransferases (MTases), MamA, MamB, and HsdM, which each target a different sequence motif for N6-adenine methylation (*Shell et al., 2013*; *Zhu et al., 2016*). Previous studies have shown that loss-of-function (inactive)

variants in these genes are common, and often associate with lineage (*Phelan et al., 2018*; *Zhu et al., 2016*). These minor differences in genotype result in radically different methylomes, potentially explaining the phenotypic variation observed between lineages (*Phelan et al., 2018*). However, these studies examined only a handful of isolates from each lineage of the *Mycobacterium tuberculosis* complex (MTBC), included few or no resistant isolates, and did not directly examine kinetics at each motif, relying instead on the motifs identified from single-molecule, real-time sequencing (SMRT-sequencing) software to classify isolates as lacking methylation of a motif entirely.

Prokaryotic DNA methylation has been shown to mediate diverse functions (*Hernday et al., 2002*; *Ringquist and Smith, 1992*; *Wright et al., 1997*), far beyond its originally understood role as the self-protective component of Restriction-Modification systems (RM systems). MTases that lack restriction enzymes, termed 'orphan' MTases (*Blow et al., 2016*) are highly conserved within phyla, and nearly ubiquitous, though diverse, across phyla, suggesting orphan MTases are functionally important (*Blow et al., 2016*). In *M. tuberculosis*, both MamA and HsdM are orphan MTases, while MamB shares sequence homology with Type IIG RM enzymes (*Zhu et al., 2016*), and therefore ostensibly has a functional restriction endonuclease domain. One emerging role for orphan MTases is regulating transcription (*Ardissone et al., 2016*; *Low and Casadesús, 2008*). In *M. tuberculosis*, MamA-mediated transcriptional regulation has been demonstrated at four promoters in *M. tuberculosis* (*Shell et al., 2013*), presumably through interaction with the −10 promoter element. A second characterized mechanism of DNA methylation mediated cis-regulation is interaction between methylated bases, MTases, and transcription factors (*Beaulaurier et al., 2015*; *Hernday et al., 2002*; *Stephenson and Brown, 2016*), which has been hypothesized to occur at specific sites in *M. tuberculosis* (*Phelan et al., 2018*; *Zhu et al., 2016*).

Previous interrogation of cis-regulation by DNA methylation in *M. tuberculosis* through SMRT-sequencing identified seven differentially methylated sites upstream of differentially expressed genes (*Gomez-Gonzalez et al., 2019*). However, only Euro-American and Indo-Oceanic (IO) isolates were analyzed, and only the 200 base pairs (bp) upstream of differentially expressed genes were examined for methylation, rather than a window more rigorously informed by mapped promoters and potential mechanisms of cis-regulation. More recently, an integrative analysis of SMRT-sequencing kinetics and differential transcription (RNAseq) following MTase knockout (Δ*hsdM*) in a *M. tuberculosis* clinical isolate concluded that HsdM-methylation did not directly affect transcript levels under their tested conditions (*Chiner-Oms et al., 2019*). However, the single tested condition and limited definition of HsdM-methylated promoters (overlap with SigA Pribnow boxes) invoke skepticism toward this conclusion. Thus, the extent of DNA methylation mediated transcriptional regulation in the MTBC and its effect on phenotype remain open questions.

Heterogeneous DNA methylation has been reported in several bacterial species (*Casadesús and Low, 2013*). Heterogeneous methylation can be caused by spontaneous mutations inactivating MTase coding genes (*Casadesús and Low, 2013*), site-specific occlusion from DNA-binding proteins (*Atack et al., 2018*; *Beaulaurier et al., 2015*; *Wallecha et al., 2002*), or intracellular stochastic methylation (*Beaulaurier et al., 2015*). When DNA methylation regulates gene expression, heterogeneous methylation creates multiple phenotypes within isogenic populations (*Casadesús and Low, 2013*). This phenotypic plasticity aids rapid adaptation to changing environmental pressures (*Cota et al., 2016*) and nutrient constraints (*Atack et al., 2018*) for other bacteria. However, no study has examined heterogeneous methylation in *M. tuberculosis*.

Here, through comparative, functional, and heterogeneity analysis, we sought to characterize how DNA adenine methylomes vary within and across MTBC clinical isolates.

## Results

### Integrative analysis of whole DNA adenine methylomes of 93 MTBC clinical isolates

We analyzed MTBC DNA adenine methylomes through four strategies (*Figure 1*). First, we used finished genomes assembled from long-read sequencing data and transferred annotations of functional and regulatory elements from virulent *M. tuberculosis* type strain H37Rv. This retained syntenic relationships and enabled facile comparative analyses without invoking assumptions inherent to *ab initio*

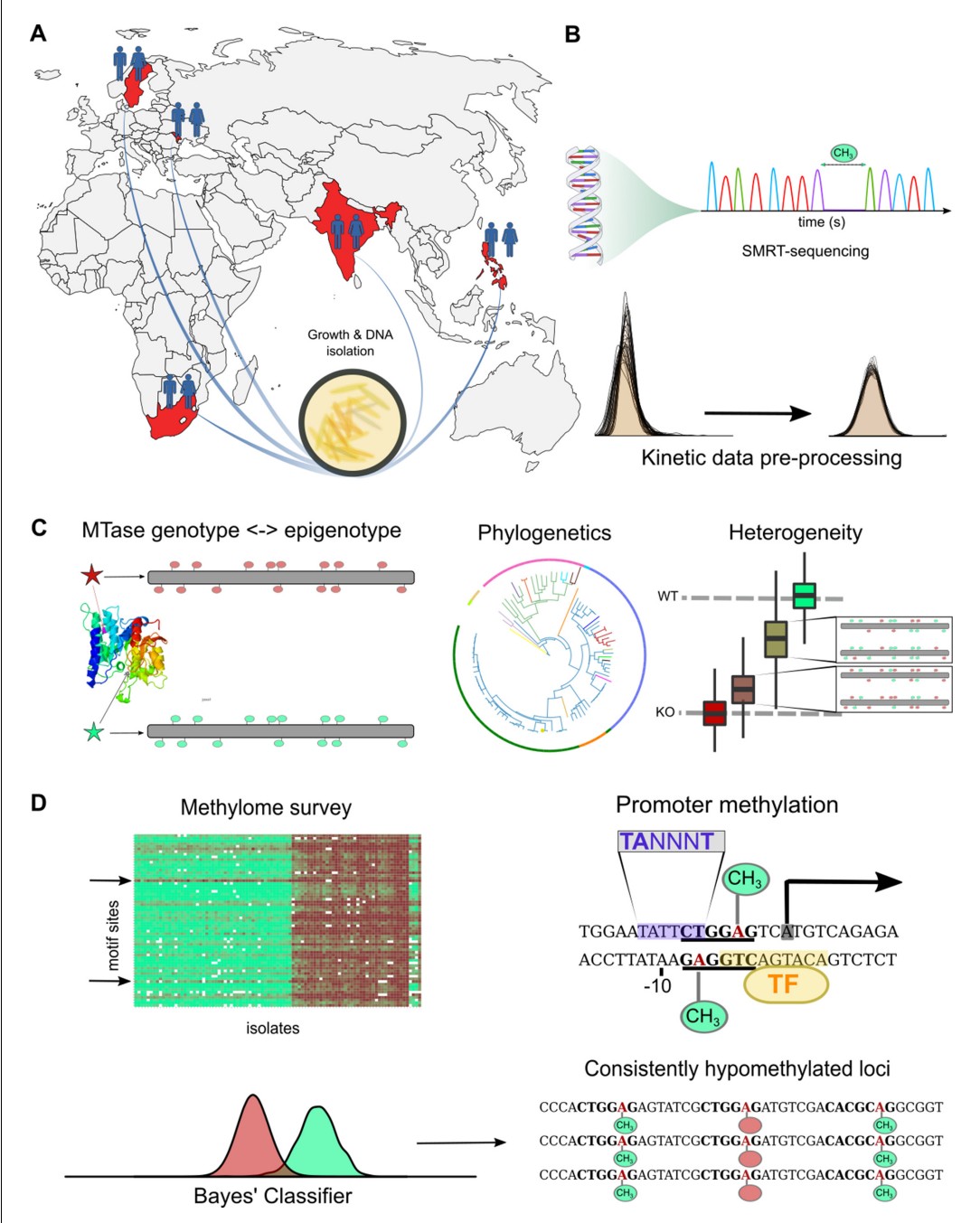

**Figure 1.** Study design and approach to whole-methylome analysis. (**A**) Isolate selection. *M. tuberculosis* clinical isolates were obtained from tuberculosis (TB) patient sputa from four countries of high TB-burden (India, Moldova, South Africa, and The Philippines), and Sweden (primarily isolated from migrants originating in high TB-burden countries). Isolates were cultured, DNA extracted and sent to the Genomic Medicine Genomics Center at UCSD for amplification-free sequencing (PacBio RSII, P6C4 chemistry). Clinical isolates were supplemented by technical replicate control runs of avirulent reference strains, and publicly available clinical isolates along with technical triplicates of H37Rv (BioProject Nos. PRJNA555636, PRJNA329548, PRJEB8783). (**B**) Methylome assembly and annotation. Raw kinetic data were log-transformed, scaled, and standardized according to run-specific statistics in unmodified bases (*Figure 1—figure supplement 1*). Variation between technical replicates were used to adjust priors based on coverage to build pdfs for a Bayesian classifier. Methylation status of characterized motifs was classified with a Bayes' classifier based on kinetics of unmodified and modified bases for each motif (Materials and methods). We processed all motifs of identified MTBC complex methyltransferases (*Zhu et al., 2016*) and assembled the methylome of each isolate with each motif site classified as methylated, hypomethylated, or indeterminate. We annotated each assembled methylome with overlapping and proximal features transferred by Rapid Annotation Transfer Tool (RATT) (*Otto et al., 2011*) from virulent type strain H37Rv. (**C**) Methylome variation. We mapped MTase genotypes to methylation levels of their motifs to describe novel

*Figure 1 continued on next page*

*Figure 1 continued*

active, partially active, and inactive alleles responsible for varying degrees of motif methylation (*Supplementary file 1*). We analyzed heterogeneity with SMALR (*Beaulaurier et al., 2015*) to characterize the capacity for methylomic variation within isogenic colonies and to probe for phase variation. We applied phylogenetic analysis of MTase genotypes and their corresponding methylation activity profiles to determine how DNA methylation across evolutionary time and identify epigenetic convergence across lineages. (D) Comparative and functional methylomics. We surveyed whole methylomes to identify motif sites and isolates with anomalous patterns. To examine motif sites consistently classified as hypomethylated for previously described causes of hypomethylation (*Beaulaurier et al., 2019*), we screened against published TFBS affinities (*Minch et al., 2015*) for interactions with DNA methylation (*Supplementary file 5*). We also screened for proximal motif sites among hypomethylated bases, which can create epigenetic 'switches' (*Hernday et al., 2002*). To identify putative cis-regulatory interactions with DNA methylation (*Supplementary file 6*), MTase motif sites were integrated with promoter element annotations (transcription start sites and sigma factor binding motifs).

The online version of this article includes the following figure supplement(s) for figure 1:

**Figure supplement 1.** Kinetic data pre-processing and quality control.

methods. Second, we included the sequencing kinetics at all MTase motif sites in every isolate. This contrasts with the approach of prior *M. tuberculosis* methylome studies (*Chiner-Oms et al., 2019*; *Gomez-Gonzalez et al., 2019*; *Phelan et al., 2018*; *Shell et al., 2013*; *Zhu et al., 2016*), which examined only the MTase motifs identified by PacBio MotifMaker (https://github.com/PacificBio-sciences/MotifMaker). This systematic approach revealed MTase genotypes with partial function, previously mischaracterized as completely inactive (*Gomez-Gonzalez et al., 2019*). Third, we used phylogenetic analysis to identify convergent MTase activity and heterogeneity analysis to characterize epigenomic diversity in vitro. Finally, we integrated transcription start sites (TSSs), transcription factor binding sites (TFBSs), and sigma factor binding sites (SFBSs) into our methylomic analyses. This helped us identify promoter methylation and potential cis (promoters and SFBSs) and trans (TFBSs) interactions between MTase motif sites and regulatory effectors.

For every isolate, the average inter-pulse duration (IPD) ratio was calculated across the reads mapping to each base. To characterize the noise in these measurements, we compared the IPD ratios at each base between technical replicates of reference strain H37Ra (*Figure 1—figure supplement 1*). We then scanned the genomes for all matches to the known target motifs (*Zhu et al., 2016*) of established *M. tuberculosis* MTases and examined their IPD ratios (*Figure 2A*). Mean single-strand coverage at target motif sites within clinical isolates ranged from 25x-152x (median = 64).

## Epigenomic convergence across lineages

Comparing the distribution of these IPD ratios in each isolate established their MTase activity profile and the functional impact of MTase mutations (*Figure 2A*, *Supplementary file 1*). Each MTase had at least three distinct loss-of-function or partially active mutations (*Figure 2A*). To determine how activity profiles distributed across *M. tuberculosis* evolution, we mapped the MTase genotypes to a phylogenetic tree (*Figure 2B*). MTase activity converged across lineage (through multiple distinct mutations) and varied within lineage (*Figure 2B and D*). The East-Asian (EAS), Euro-American, and IO lineages each had multiple profiles among their members (*Figure 2B and D*). This contradicts one previous analysis on a smaller isolate set that reported lineage-specific methylation (*Phelan et al., 2018*), and corroborates the opposing conclusion reached in another recent publication (*Chiner-Oms et al., 2019*).

## Diverse mutations drive DNA methyltransferase activity profiles

Cumulatively, the 93 *M. tuberculosis* and *M. africanum* isolates harbored 40 distinct mutations within the known MTase genes *mamA*, *mamB*, and *hsdM/hsdS*, including 29 previously unreported (*Supplementary file 1*). Comparing the IPD ratio at each base matching the MTase target motifs across isolates revealed the effect of each variant on MTase activity (*Figure 2A*). The isolates had four novel loss-of-function mutations: *mamB* H770N, *mamB* DELG2543, *mamB* INS1181-1583, and *hsdM* DEL900-909. This comparison also confirmed that variant *hsdS* L119R inactivates MTase activity in the HsdM complex. Previously, *hsdS* L119R had been observed only in tandem with another loss-of-function mutation, *hsdM* G173D (*Chiner-Oms et al., 2019*; *Phelan et al., 2018*). One isolate had *hsdS* L119R alone, and its HsdM motifs were unmodified (*Figure 2A*). Twenty-nine MTase mutations had no apparent effect on MTase activity, 23 of which were previously unreported

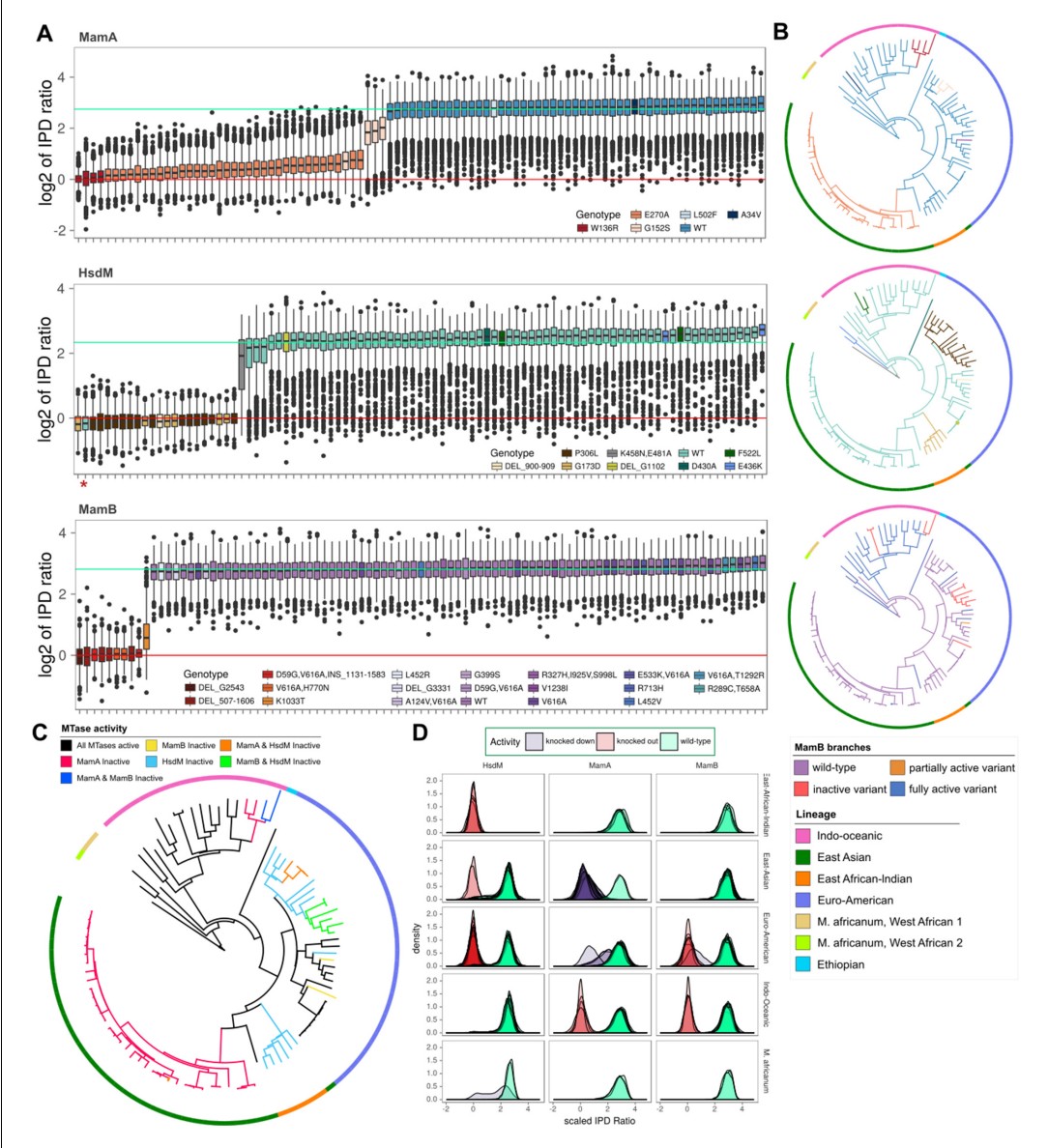

**Figure 2.** MTase activity patterns and genotypes across clinical and reference strains. (A), Boxplots of inter-pulse duration (IPD) ratio distributions within MamA (top pane), HsdM (middle pane), and MamB (bottom pane) target motifs for each *M. tuberculosis* or *M. africanum* isolate. Boxplots are colored by *mamA*, *hsdM*, and *mamB* genotype. In one isolate, *hsdS* L119S mutation disrupted methylation capacity and is marked by an asterisk (*). In each pane, horizontal lines mark the mean $\log_2$ (IPD ratio) of motif sites for isolates with active (mint line) and inactive (red line) MTases. (B) SNP-based phylogenetic trees with mutations mapped for each MTase. Isolates are colored by MTase genotype using the same colors as the boxplots in A, except for MamB, which is colored by MTase activity. The phylogeny was built using maximum likelihood on a concatenation of 22,393 SNPs with *M. bovis* and *M. canetti* as outgroups. Colors of the outer rung indicate lineage. (C) Phylogeny of isolates in this study with branches colored according to the MTase activity profile. Colors of the outer rung indicate lineage according to the same color scheme as B. (D) Density traces of sequencing kinetics for each isolate at every motif site, organized into panes by MTase (columns) and lineage (rows), and colored by the activity of their MTase. The fill underneath each density trace is rendered translucent such that overlapping distributions from multiple isolates appear progressively darker as a function of the number of overlapping distributions.

The online version of this article includes the following source data and figure supplement(s) for figure 2:

**Source data 1.** Scaled $\log_2$ (IPD ratios) for each site with MTase genotype.
**Source data 2.** Scaled $\log_2$ (IPD ratios) for each site across isolates with lineage metadata.
**Figure supplement 1.** Correction of a mischaracterized *mamB* variant effect via direct comparative genomics.
**Figure supplement 2.** Methyltransferase activity by isolate count.

(*Supplementary file 1*). All isolates had at least one active MTase (*Figure 2C*), yet only 32/93 clinical isolates had all three MTases active (*Figure 2—figure supplement 2*).

In total, 12 different mutations had reduced or inactive MTase function (three *hsdM/hsdS*, five *mamB*, and four *mamA*), demonstrating that MTase disrupting mutations are repeatedly selected for, although almost all individual mutations are monophyletic (*Figure 2A and B*).

Recently, *mamB* D59G was reported as the sole variant in a MamB inactive isolate (*Chiner-Oms et al., 2019*) (SRA: ERR956955, ERR964401). We identified four Indo-Oceanic isolates with *mamB* variant D59G, but all four also harbored V616A (*Supplementary file 2*). Two of these four isolates were active and had no other mutations. However, the other two were MamB inactive and carried a 1356 bp insertion that we have identified as an IS6110 insertion sequence (*Figure 2—figure supplement 1*). One of these inactive isolates was the same isolate recently reported with *mamB* D59G alone (*Chiner-Oms et al., 2019*). The prior study did not report the *mamB* insertion, likely due to their reference-mapping of short reads to call MTase variants (*van Dijk et al., 2018*). However, it is unclear why their methods did not capture V616A. Considering that MamB was active in isolates carrying *mamB* D59G without the insertion, we conclude the insertion was responsible for inactivating MamB.

Our approach resolves the disputed functional impact of *mamA* variant E270A ($mamA_{E270A}$), common to the EAS lineage (ref (*Phelan et al., 2018*) and *Supplementary file 2*). Nucleoside digestion previously showed low-level methylation in $mamA_{E270A}$ mutants (*Shell et al., 2013*). However, subsequent SMRT-sequencing studies reported contradictory findings. Three studies reported a lack of methylation at all MamA motif sites in $mamA_{E270A}$ isolates (*Gomez-Gonzalez et al., 2019*; *Phelan et al., 2018*; *Zhu et al., 2016*) while another reported partial methylation in one $mamA_{E270A}$ isolate, and a complete lack of methylation in two $mamA_{E270A}$ isolates (*Chiner-Oms et al., 2019*). In the 34 isolates with $mamA_{E270A}$, MamA motif sites had intermediate IPD ratios (*Figure 2A*, top), indicating partial activity. Our analysis revealed three additional MTase alleles that conferred partial MTase activity: $hsdM_{K458N;E481A}$, $mamA_{G152S}$, and $mamB_{K1033T}$ (*Figure 2A*). These isolates had IPD ratio distributions consistent with neither homogenously methylated nor unmethylated sites (*Figure 2A & D*).

We hypothesized that these isolates' intermediate IPD ratios were due to heterogeneous methylation. IPD ratios are reported as the average IPD ratio observed across sequencing reads mapped to each position, which originate from different cells. Therefore, if isolate colonies contained subpopulations of cells with different methylomes, it would result in the intermediate IPD ratios we observed. Heterogeneous methylation can be caused by different phenomena, including phase-variant MTase activity, intracellular stochastic methylation, or transient hemimethylation from a subset of cells undergoing cell division. To distinguish between these possibilities, we examined IPD signals at the read level using SMALR (*Beaulaurier et al., 2015*).

## A subset of DNA MTase alleles drive constitutive intercellular mosaic methylation in *M. tuberculosis*

Within each sequencing read, the software SMALR (*Beaulaurier et al., 2015*) calculates the average IPD (natural log-transformed) across multiple motif sites targeted by the same MTase. This 'native IPD value' will be higher in reads where each motif site is methylated, lowest in reads with no motif sites methylated, and intermediate in reads with a fraction of motif sites methylated (*Figure 3H*). Thus, the distribution of native IPD values in an isolate can discriminate homogenous methylation (or unmethylation) from two types of heterogeneous methylation: phase-variant MTase activity and intracellular stochastic methylation (*Beaulaurier et al., 2015*). Phase-variant MTase activity causes a distinctive bimodal distribution in SMALR native IPD values, while intracellular stochastic methylation results in a unimodal distribution with a mean between that of fully methylated and fully unmethylated sequences (*Beaulaurier et al., 2015*). SMALR native IPD values can also potentially distinguish transient hemimethylation from intracellular stochastic methylation and phase-variant MTase activity. Previously, transient hemimethylation was attributed to the small fraction (0.07%) of unmethylated reads observed in cultures with otherwise fully methylated reads (*Beaulaurier et al., 2015*). Transient hemimethylation would thus likely manifest in SMALR native IPD values as a bimodal distribution with a small minority of entirely unmethylated reads among a majority of methylated reads.

The native IPD values for MamA motif sites in each sequencing read distributed normally in the 49 *mamA* wild-type isolates, with a mean native IPD of 2.15 (per-isolate means: 1.96–2.28)

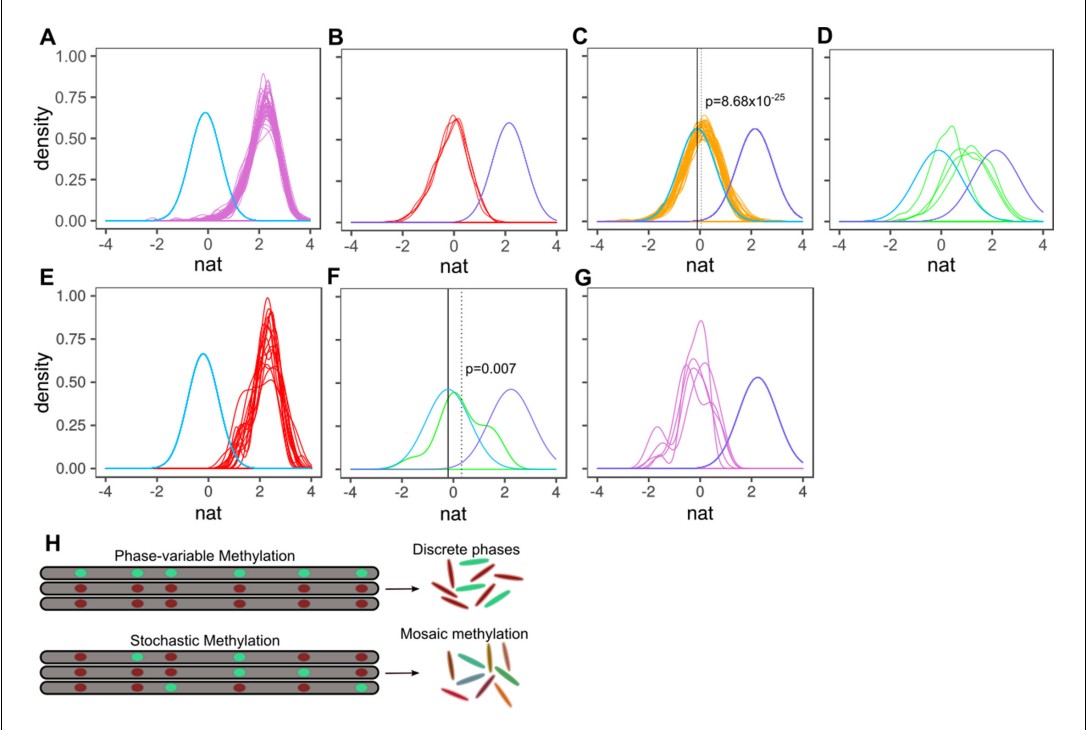

**Figure 3.** Characterizing methylation heterogeneity in *M.tuberculosis* clinical isolates. Native IPD value (nat) is the subread-mean normalized natural log of IPDs (as output from SMALR *Beaulaurier et al., 2015*) across all motif sites within a subread. (A-G) depict the distribution of mean native IPD values among subreads for each isolate of the specified MamA or MamB allele. Each colored trace corresponds an isolate. Each pane has two reference curves: a mean native IPD value, number of measurements (isolates), and identical standard deviation identical to those of the inactive genotypes (light blue) and a curve with a mean native IPD value, standard deviation, and number of observations (isolates) identical to that of isolates with wild-type MamA (A-D) or MamB (E-G) activity (light violet). (A) *mamA*~WT~. (B) *mamA*~W136R~. (C) *mamA*~E270A~. The dotted vertical line marks the mean native IPD value for all *mamA*~E270A~ isolates. (D) *mamA*~G152S~. (E-G) Putative heterogeneous methylation by *mamB*~K1033T~ mutant (same as A-D, but for *mamB*). Only isolates with >19 qualifying subreads are included (at least 6 CACGCAG motifs are required within the subread to qualify). (E) Wild-type MamB (n = 16 passing inclusion criteria). There is no consistent signature of stochastic heterogeneity or phase variation. (F) *mamB*~K1033T~ (n = 1) appears to exhibit low MTase activity, with mean native IPD dotted line well below that of wild-type MamB and the inactive genotypes (solid black line) and significantly greater than the mean native IPD value of inactive genotypes, suggesting stochastic heterogeneity. This native IPD value resembles the low-level MTase activity in isolates harboring the *mamA*~E270A~ allele. (G) All qualifying isolates (n = 5) with *mamB* inactive alleles. Despite harboring different nonsynonymous *mamB* mutations (n = 4), *mamB* motifs appear entirely unmethylated, with a native IPD value consistent with inactive MTase genotype (*Beaulaurier et al., 2015*) and no clear signals of phase variation or intracellular stochastic heterogeneity. (H) Stochastic versus phase-variable methylation. Conceptual illustration depicting the distinction between methylome diversity within colonies exhibiting phase-variable methylation (top) and stochastic methylation (bottom). Each gray segment represents a chromosome from an individual cell within the colony. Each oval within the segment represents a motif site, illustrated as methylated (mint) or unmethylated (red).

The online version of this article includes the following source data and figure supplement(s) for figure 3:

**Source data 1.** Read-level sequencing kinetics of MamA MTase motif sites in 93 MTBC clinical isolates.
**Source data 2.** Read-level sequencing kinetics of MamB MTase motif sites in 93 MTBC clinical isolates.
**Figure supplement 1.** Observed distribution of native IPD values in methionine-starved H37Rv~ΔmetA~ compared to simulated phase-variant mixture.

(*Figure 3A*), indicating that MamA motifs on most reads in these isolates were entirely methylated (*Beaulaurier et al., 2015*). The four isolates with inactive allele *mamA*~W136R~ each distributed normally (mean = −0.107, *Figure 3B*) indicating a consistent lack of MamA-methylation. In contrast, the four isolates with partially active variant *mamA*~G152S~ distributed normally with varied per-isolate means (range: 0.22-0-1.19). Their consensus mean native IPD value (0.766) was substantially above *mamA*~W136R~ (Cohen's d = 1.10, 95% CI 1.04–1.17; two-tailed t-test p = $3.20 \times 10^{-207}$) and below *mamA* wild-type isolates (d = 2.14, CI 2.09–2.20, p<$2.23 \times 10^{-308}$, *Figure 3C*, *Figure 3—source data 1*). Each *mamA*~G152S~ isolate individually had a lower mean value than wild-type as well (d for isolate with lowest effect size = 0.48, CI 0.39–0.57, p = $9.74 \times 10^{-21}$). This unimodal, normal distribution of intermediate native IPD values implies intracellular stochastic methylation, rather than phase-variant

MTase activity (*Beaulaurier et al., 2015*). While phase-variant MTase activity has been observed in many bacteria, intracellular stochastic methylation has previously only been observed in a single species, *Chromohalobacter salexigens* (*Beaulaurier et al., 2015*).

After observing this intracellular stochastic methylation in $mamA_{G152S}$ isolates, we analyzed heterogeneity in isolates with partially active MamA (E270A) and MamB (K1033T). The 34 $mamA_{E270A}$ isolates had a methylated fraction (mean = 0.0558, *Figure 3C*) lower than those of wild-type isolates (mean = 2.15) and higher than the methylated fraction of isolates with inactive allele W136R (d = 0.23, p = $8.68 \times 10^{-25}$), indicating intracellular stochastic methylation. Native IPD values for *mamB* wild-type (mean = 2.24) were significantly (p = 0.0006) though slightly (d = 0.15, CI 0.05–0.25) higher than for *mamA* wild-type, consistent with homogenous methylation. MamB inactive alleles (mean = −0.235) were modestly lower (d = 0.19, CI 0.008–0.377) than inactive allele MamA$_{W136R}$ with borderline statistical significance (p = 0.041), suggesting that $mamA_{W136R}$ may have retained a miniscule amount of MTase activity. However, the effect of methylation of sequencing kinetics can vary between motif sequences, and so we cannot conclude from our data that limited activity is present. We therefore continue to consider m$amA_{W136R}$ to be an inactive allele. Like $mamA_{E270A}$, the mean native IPD value for $mamB_{K1033T}$ was greater than in inactive alleles (d = 0.77, CI 0.34–1.21, p = 0.007) consistent with intracellular stochastic methylation (*Figure 3E–G*). Multiple HsdM motif sites clustered together on the same subread too infrequently for analysis with SMALR, precluding analysis of potential heterogeneity with SMALR (isolates had on average 3898.6 MamA, 822.9 MamB, and 719.4 HsdM sites).

While these intermediate native IPD value distributions for $mamA_{E270A}$, $mamA_{G152S}$, and $mamB_{K1033T}$ are consistent with intracellular stochastic methylation, they would also be produced by consistent methylation of some motif sites and consistent lack of methylation in the remaining sites. By examining isolates with these partially active alleles, we see that mean IPD ratios for motif sites distribute unimodally at intermediate values rather than bimodally (purple-filled traces for MamA and MamB, *Figure 2D*). Therefore, we reject this alternative explanation and conclude that intracellular stochastic methylation generates diverse combinations of methylated motif sites within the bacilli from which the sequenced DNA was isolated. We call this epigenetic mosaicism 'intercellular mosaic methylation'. The methylomic diversity in intercellular mosaic methylation is distinct from the two discrete phases generated from phase-variable methylation, instead generating a mosaic of methylation patterns among the cells comprising an isogenic colony (*Figure 3H*).

Next, we asked whether methionine restriction could induce intercellular mosaic methylation in isolates with wild-type MTase function. Methionine is directly linked to DNA methylation by its requirement of S-adenosyl methionine (SAM) as the methyl group donor (*Parveen and Cornell, 2011*). We reasoned that if methionine starvation induces intercellular mosaic methylation, the effect could be extrapolated to other nutrient restrictions that limit flux through the adenine methylation reaction, such as precursor metabolites for SAM synthesis or trace metal ion cofactors required for SAM biosynthesis (*Czyrko et al., 2018*) and adenine methylation (*Bist and Rao, 2003*). We ran SMALR on published kinetic data from *metA*-knockout H37Rv methionine auxotrophs (Δ*metA*) SMRT-sequenced following 5 days of methionine starvation (*Berney et al., 2015*).

To test for intercellular mosaic methylation in Δ*metA* we compared its native IPD value distribution to a mixture of wholly methylated and wholly unmethylated reads (*Figure 3—figure supplement 1A*). This simulated mixture was sampled from *mamA* wild-type and inactive isolates, in proportion to produce the same mean as Δ*metA*. The Δ*metA* native IPD value distribution was distinct from the simulated mixture, with fewer fully methylated reads and more partially methylated reads (*Figure 3—figure supplement 1*). These results are consistent with a mixture of fully methylated reads inherited from bacilli born prior to methionine deprivation and stochastically methylated reads from daughter strands that underwent partial re-methylation following starvation. These results suggest that intercellular mosaic methylation can be induced by nutrient limitation (at least for methionine, and presumably other nutrient constraints that limit flux through the adenine methyltransferase reaction), providing a potential mechanism of environmentally induced phenotypic heterogeneity.

## Anomalous methylation patterns in orphan MTase motif sites

Next, we surveyed all common motif sites (present in ≥75 isolates, n = 4,486; *Supplementary file 3*) for patterns in their variation across isolates and motif sites. IPD ratio distributions were stable within

isolates the same MTase allele except for partially active mutants (*Figure 4A*), consistent with prior observations that motif sites are typically invariably methylated or invariably unmethylated (*Blow et al., 2016*). Most of the isolates with anomalous methylation had *mamA* alleles with partial activity, while a few isolates with *hsdM* alleles with full activity had anomalous methylation patterns (arrows, *Figure 4B*).

Analyzing variation across motif sites of active wild-type isolates revealed three interesting features. First, while most motif sites were primarily methylated (light yellow), a subset had significantly lower median IPD (*Figure 4A–C*). Second, among the mostly methylated motif sites, a handful had lower IPDs in a subset of isolates (*Figure 4A–C*, *Figure 4—figure supplement 2B*). The converse was also true; some motif sites with low median IPDs had higher IPDs in a subset of isolates. Third, partially active MTases had distinct methylation profiles with median IPDs between those of inactive and wild-type (*Figure 4D*). All three features were more pronounced in HsdM and MamA motif sites than in MamB (*Figure 4A–C,E*).

MamA and HsdM are orphan MTases, which frees their motif sites to remain unmethylated without risking degradation from a cognate restriction enzyme (*Blow et al., 2016*), potentially explaining their variable methylation (*Zhu et al., 2016*) compared to MamB. To corroborate MamB as a full RM system rather than an orphan MTase, we searched InterPro database for functional domains and identified a putative restriction endonuclease (REase) domain (*Figure 4—figure supplement 1*) as its cognate REase. Single chain restriction-modification fusions are a defining characteristic of Type IIG RM systems (*Shen et al., 2011*), which MamB belongs to *Zhu et al., 2016*. Orphaned MTases can safely leave motif sites unmethylated, affording the opportunity to evolve essential functions (*Blow et al., 2016*). At such sites, advantageous methylation states can be selected for following prolonged culturing. Considering this, we sought to identify motif sites that were hypervariable across strains with active MTases, reasoning that it would indicate differential selection for methylation status during culture prior to sequencing and highlight interesting differences between strains. To find hypervariable sites, we calculated the standard deviation (SD) in IPD ratio across isolates for each MTase site. Variation across isolates for each qualifying motif site was compared against the distribution of SD (standard deviation of SD across MamB sites) in MamB sites (*Figure 4E*, *Figure 4—figure supplement 2C*), because they distributed normally and with few outliers (*Figure 4—figure supplement 2A*). We defined motif sites with SD ≥3 standard deviations above the mean SD of MamB motif sites. Of 4486 common (in ≥75 isolates) MTase motif sites, 351 were hypervariable (*Supplementary file 4*). These hypervariable sites fell within 204 coding regions, and within promoter-proximal regions upstream (within 100 bp) of 25 TSSs (*Supplementary file 4*). Yet only seven hypervariable sites were MamB motifs. Hypervariability was 9.43-fold more prevalent in motif sites of orphan MTases than in MamB motif sites (95% CI: 4.50–23.7). This marked enrichment (p = $3.39 \times 10^{-17}$, two-tailed Fisher's exact test) of hypervariable motif sites in MamA (n = 240) and HsdM (n = 104) is consistent with the lack of selection pressure against unmethylated sites characteristic of orphan MTases (*Blow et al., 2016*).

## Hypomethylated MTase motif sites are rare yet remarkably consistent across isolates

One characteristic of orphan MTases is hypomethylated sites, specific genome sites lacking or absent of methylation despite the presence of an MTase target motif and active MTase (*Blow et al., 2016*). Hypomethylated sites have previously been found in *M. tuberculosis* (*Zhu et al., 2016*) and many other bacteria (*Blow et al., 2016*; *Hernday et al., 2002*; *Ringquist and Smith, 1992*). To identify hypomethylated sites in each of our isolates, we applied Bayesian classification to their sequencing kinetics. Active isolates averaged 20.7 hypomethylated HsdM sites, 13.4 hypomethylated MamA sites, and 0.289 hypomethylated MamB sites (*Figure 5—figure supplement 1*), comparable to previous reports (*Zhu et al., 2016*). However, while rare, these hypomethylated motif sites showed remarkable consistency, with the same MTase motif loci hypomethylated in multiple isolates. The most conserved hypomethylated locus was within *mmpL4*, 1,719 bp downstream of its start. Despite having a MamA target motif, this locus was unmethylated in 51 MamA active isolates (*Table 1*). In total, 34 MamA loci and 58 HsdM loci were consistently hypomethylated (p-value<4.72E-07, cumulative binomial, *Supplementary file 5*). These loci included 18 of the top 10 frequently hypomethylated MamA and HsdM sites in a previous study of 12 MTBC strains (*Table 1*). This consistency

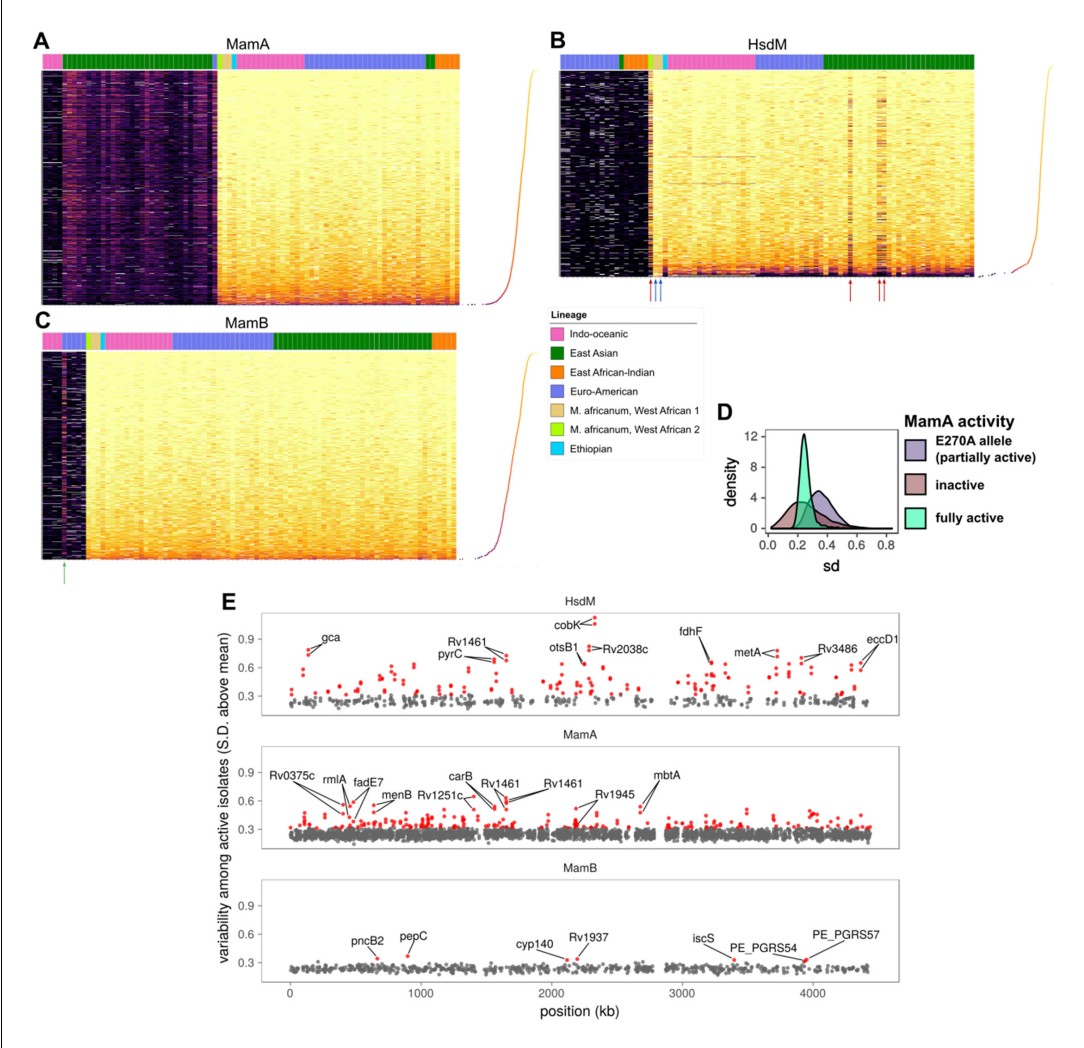

**Figure 4.** Comparative methylomics identifies DNA methylation anomalies in orphan MTases and motif sites. Heat maps of sequencing kinetics for (**A**) MamA, (**B**) HsdM, and (**C**) MamB motifs at all common (in ≥75 isolates) motif sites (y-axis), descending according to median sequencing kinetics (log$_2$ (IPD ratio)). Isolates (x-axis) are sorted from left to right by activity level, lineage, and genotype in decreasing priority. Lineages are Indo-Oceanic (IO), East-Asian (EAS), East-African-Indian (EAI), Euro-American (E), Ethiopian (Lineage 7), and the *M. africanum* lineages 5 and 6. Dots on the rotated plot adjacent to the heatmap express the median log$_2$ (IPD Ratio) for each site across isolates. Darker and lower dots indicate a lower median (log$_2$ (IPD Ratio)). Red arrows mark isolates with wild-type or near wild-type MTase activity that exhibit hypomethylation at more motif sites (dark bands) than other wild-type isolates. Blue arrows mark two isolates with significantly fewer hypomethylated motif sites than other isolates with wild-type HsdM activity, for unknown reasons. The green arrow in the MamB plot marks an isolate with an IPD ratio significantly higher than expected for an inactive isolate, suggesting some MTase activity for the genotype. (**D**) Distribution of standard deviation (sd) sizes among MamA motif sites across isolates with one of three methylation activity levels: partial MamA activity from the E270A mutation common to EAS isolates, the loss-of-function W136R mutation ('knockout mutant'), or one of the genotypes encoding MamA with wild-type methylation activity. (**E**) Position (x-axis) in a representative genome and variability sd of (log$_2$ (IPD Ratio)) in sequencing kinetics across isolates with active MTase (y-axis) at common motif sites (present in 75 or more isolates). Motif sites within three sd of the mean for MamB motifs are gray, and the outliers (>3 sd from mean) are highlighted in red. CDSs within which each of the top 10 most variable sites for each MTase occur are labeled, along with their palindromic partner motif site.

The online version of this article includes the following source data and figure supplement(s) for figure 4:

**Source data 1.** Sequencing kinetics for all common (in >75 isolates) MamA motifs.
**Source data 2.** Sequencing kinetics for all common (in >75 isolates) HsdM motifs.
**Source data 3.** Sequencing kinetics for all common (in >75 isolates) MamB motifs.
**Source data 4.** Per-site sequencing kinetics variability for all common (in >75 isolates) MamA motifs, with MamA activity.
**Source data 5.** Per-site sequencing kinetics variability for all common (in >75 isolates) motifs, with variability classification.
**Figure supplement 1.** *mamB* mutations mapped to InterPro functional domains and predicted 3D structure.
**Figure supplement 2.** Statistical features of hypomethylated and hypervariable motif sites.

**Table 1.** Consistently Hypomethylated MTase Motif Sites Across Clinical *M.tuberculosis* isolates.

The top 20 most significant hypomethylated loci from each MTase. For each methyltransferase ('MTase') motif target locus ('Gene', 'Sense', and 'Position'), we counted the number of isolates in which the isolate was hypomethylated and the total number of isolates that possessed the locus ('Hypomethylated'). This fraction was used to perform a cumulative binomial probability test ('p-value'). Loci with p-values below 4.72E-07 were considered significant at 0.01 significance level, after Bonferroni correction for multiple hypothesis testing. Loci were assigned by our methylome annotation pipeline using H37Rv reference annotations transferred from Rapid Annotation Transfer Tool (RATT) (*Otto et al., 2011*). For each palindromic pair, the locus with the most significant hypomethylated fraction is reported. In case of a tie, the locus on the same strand as the gene is reported. The fraction of active isolates hypomethylated at the partner site is included ('Palindrome'). The surrounding 20 bases of each loci were scanned for transcription factor binding site motifs previously characterized in *M. tuberculosis* (*Minch et al., 2015*). The most significant motif match was included ('Top TF'). Only transcription factor binding motifs with an E-value below 0.01 were scanned for, and only matches with a p-value (converted log-likelihood ratio score) below 0.0001 were reported. MTase motif loci less than 100 bp from another locus targeted by the same MTase were labeled ('Yes' in column 'Nearby Motif'). Genes that were previously reported (*Zhu et al., 2016*) to contain frequently hypomethylated sites are marked with an asterisk.

| Gene | Sense | Position | Hypomethylated | p-value | Palin-drome | Top TF | Annotated Function | Nearby Motif |
|------|-------|----------|----------------|---------|-------------|--------|--------------------|--------------|
| **HsdM** | | | | | | | | |
| rocA* | sense | 834 | 67/70 | 2.49E-99 | 67/70 | Rv3488 | Probable pyrroline-5-carboxylate dehydrogenase | |
| cobK* | sense | 304 | 50/68 | 6.15E-62 | 50/68 | Rv2788 | Precorrin-6X reductase | Yes |
| Rv1461* | antisense | 559 | 47/70 | 3.17E-55 | 37/70 | Rv1956 | Iron-sulfur cluster assembly protein | |
| Rv2963* | sense | 683 | 46/70 | 2.10E-53 | 46/70 | Rv2788 | putative ion transporter | |
| PPE24* | antisense | 2275 | 33/33 | 1.31E-51 | 32/33 | Rv3133c | PPE family protein PPE24 | |
| metA* | antisense | 391 | 42/70 | 2.20E-46 | 40/70 | Rv2324 | homoserine O-acetyltransferase | |
| pks6* | sense | 423 | 26/34 | 1.18E-33 | 9/34 | Rv1719 | Probable membrane bound polyketide synthase | |
| PPE24* | antisense | 1807 | 17/17 | 6.14E-27 | 16/17 | Rv3133c | PPE family protein PPE24 | |
| pks6* | sense | 424 | 21/32 | 3.97E-25 | 14/32 | Rv1719 | Probable membrane bound polyketide synthase | |
| pyrC | sense | 744 | 20/70 | 5.90E-15 | 19/70 | Rv1049 | Probable dihydroorotase PyrC | |
| Rv3179 | upstream | 36 | 9/9 | 1.33E-14 | 9/9 | Rv1816 | Conserved protein | Yes |
| Rv3179 | upstream | 49 | 9/9 | 1.33E-14 | 9/9 | Rv1816 | Conserved protein | Yes |
| gcA* | antisense | 467 | 19/69 | 5.89E-14 | 14/69 | Rv1473A | Possible GDP-mannose 4,6-dehydratase | |
| Rv2279 | antisense | 693 | 10/21 | 1.01E-10 | 9/21 | Rv1776c | Probable transposase | |
| lpqG | upstream | 93 | 14/54 | 2.85E-10 | 8/54 | Rv1219c | Probable conserved lipoprotein LpqG | |
| Rv2038c | sense | 183 | 14/69 | 9.05E-09 | 9/69 | Rv2989 | Probable sugar-transport ATP-binding protein ABC transporter | |
| PPE24* | sense | 1584 | 5/5 | 1.95E-08 | 5/5 | Rv0818 | PPE family protein PPE24 | |
| **MamA** | | | | | | | | |
| mmpL4* | sense | 1719 | 51/51 | 2.18E-126 | 49/51 | Rv0678 | transmembrane transport protein | |
| Rv1049 | upstream | 7 | 39/51 | 1.23E-85 | 9/51 | | Oxidation-sensing regulator MosR | |
| Rv1461* | antisense | 472 | 38/50 | 2.73E-83 | 35/50 | | Iron-sulfur cluster assembly protein Suf | Yes |
| treZ* | antisense | 1272 | 31/35 | 2.14E-72 | 26/35 | | Maltooligosyltrehalose trehalohydrolase | |
| accE5 | downstream | 447 | 20/33 | 2.89E-41 | 3/33 | | acetyl-/propionyl-coenzyme A carboxylase | |
| PPE34* | sense | 1664 | 13/20 | 7.05E-28 | 7/20 | | PPE family protein PPE34 | |
| mptA* | sense | 23 | 14/31 | 8.02E-27 | 3/31 | | Alpha(1 ->6) mannosyltransferase | |
| accA1* | sense | 598 | 13/25 | 4.65E-26 | 6/25 | Rv0339c | Probable acetyl-/propionyl-coenzyme A carboxylase alpha chain | Yes |
| Rv3282* | antisense | 325 | 12/18 | 4.93E-26 | 0/18 | | putative nucleoside-triphosphate diphosphatase | |
| fadE7* | sense | 1069 | 13/43 | 3.09E-22 | 8/43 | Rv3488 | Acyl-CoA dehydrogenase FadE7 | Yes |
| pks9* | sense | 2728 | 13/50 | 2.93E-21 | 3/50 | Rv0023 | polyketide synthase Pks9 | |

*Table 1 continued on next page*

*Table 1 continued*

| Gene | Sense | Position | Hypomethylated | p-value | Palin-drome | Top TF | Annotated Function | Nearby Motif |
|------|-------|----------|----------------|---------|-------------|--------|--------------------|--------------|
| bioB | sense | 796 | 11/49 | 2.04E-17 | 0/49 | Rv1049 | biotin synthetase | |
| treZ* | upstream | 880 | 8/13 | 2.46E-17 | 5/13 | | Maltooligosyltrehalose trehalohydrolase | |
| Rv1461* | antisense | 416 | 10/50 | 2.08E-15 | 2/50 | | Iron-sulfur cluster assembly protein | Yes |
| Rv1278 | sense | 1469 | 9/44 | 4.25E-14 | 1/44 | | Putative transport protein | |
| aldA | antisense | 642 | 8/48 | 6.49E-12 | 0/48 | Rv3830c | Probable NAD-dependent aldehyde dehydrogenase AldA | |
| Rv0370c | antisense | 389 | 8/50 | 9.17E-12 | 0/50 | | Possible oxidoreductase | Yes |
| frdA* | antisense | 787 | 8/51 | 1.08E-11 | 0/51 | | Probable fumarate reductase FrdA | |
| PPE34* | sense | 3454 | 4/4 | 1.39E-10 | 2/4 | | PPE family protein PPE34 | |
| Rv1251c | antisense | 1309 | 7/46 | 2.69E-10 | 1/46 | | Putative ester hydrolase | |
| **MamB** | | | | | | | | |
| PE_PGRS54 | sense | 112 | 10/68 | 8.16E-24 | N/A | Rv0767c | PE-PGRS family protein PE_PGRS54 | |
| PE_PGRS57 | sense | 112 | 9/68 | 3.94E-21 | N/A | Rv0767c | PE-PGRS family protein PE_PGRS57 | |

The online version of this article includes the following source data for Table 1:

Source data 1. Multisequence fasta file with context sequence of each consistently hypomethylated MTase Motif Locus.

Source data 2. Output file from FIMO, run with the fasta file *Source data 1*, and the probability weight matrices of TFBS motifs characterized in H37Rv (*Minch et al., 2015*).

would be unlikely if hypomethylation occurred randomly, suggesting there are conserved mechanisms preventing methylation at these loci.

MamB motif sites, in contrast, were thoroughly methylated in MamB active isolates (*Figure 5—figure supplement 1C*). Only two MamB motif loci were hypomethylated in a significant number of isolates, and even these loci were methylated in most isolates (*Table 1*). The near uniform methylation of MamB target sites is likely to prevent DNA cleavage from its putative cognate restriction endonuclease (*Figure 4—figure supplement 1*).

MamA and HsdM are palindromic, each targeting both a motif and its reverse complement (*Zhu et al., 2016*). Thus, every adenine methylated by these enzymes comes with a potentially methylated partner on the opposite strand. HsdM palindromic pairs were mostly hypomethylated together. In contrast, many MamA motif pairs were hemimethylated, with one site consistently hypomethylated and the other frequently methylated (*Table 1*). Site-specific hemimethylation functions as part of multiple characterized processes in other bacterial species (*Braun and Wright, 1986; Roberts et al., 1985*) but what role it serves in *M. tuberculosis,* if any, remains unknown.

## Sequence contexts of most hypomethylated sites are consistent with transcription factor occlusion

In other bacteria, hypomethylation has been attributed to transcription factors occluding MTase access to DNA when their respective target motifs overlap (Beaulaurier et al.; *Hernday et al., 2002; Stephenson and Brown, 2016*). To determine if this may be the case in *M. tuberculosis*, we scanned the context sequence of each consistently hypomethylated site for TFBS motifs previously characterized in *M. tuberculosis* (*Minch et al., 2015*). All 58 consistently hypomethylated HsdM loci matched at least one significant TFBS motif (p-value<0.0001, converted log-likelihood ratio score), suggesting transcription factor occlusion was responsible. In contrast, only 14 of the 34 consistently hypomethylated MamA loci significantly matched a TFBS motif (p-value<0.0001, converted log-likelihood ratio score; *Table 1*). The abundance of TFBS matches at HsdM motif loci may be due to the lower stringency of its motif (HsdM: GATNNNNRTAC, MamA: CTGGAG). Notably, the TFBS motif of oxidation-sensing regulator *mosR* (*Rv1049*) (*Peterson et al., 2014*) matched multiple hypomethylated MamA and HsdM loci, and the *mosR* gene itself had a hypomethylated MamA locus 7 bp upstream of its TSS (*Table 1*).

One particularly intriguing example of site-specific hypomethylation was *cobK* 304, the HsdM motif site 304 bp inside the gene *cobK*. This locus was hypomethylated in 50/68 HsdM-active isolates (*Table 1*). The IPD ratio at *cobK* 304 across HsdM-active isolates was bimodal (*Figure 5A*), supporting this finding. The 18 *cobK* 304 methylated isolates were all IO and grouped together in our phylogenetic tree (*Figure 5B*). The context sequence around *cobK* 304 matched the binding site motif of transcription factor *mntR* (*Rv2788*) (q-value = 0.0136, *Table 1*), a manganese dependent repressor (*Pandey et al., 2015*). This could explain the 50 *cobK* 304 hypomethylated isolates. If *mntR* bound to that site, it could prevent HsdM from methylating it. Meanwhile, genotyping *mntR* revealed that all 18 IO isolates shared the variant *mntR* Q131* (*Figure 5B*), a nonsense mutation found previously in IO isolates (*Gomez-Gonzalez et al., 2019*) that truncates MntR. This truncation could explain why *cobK* 304 was methylated in all 18 IO isolates.

Close proximity between MTase motif sites can also cause bacterial hypomethylation (*Casadesús and Low, 2013*), wherein methylation of one site is negatively associated with the other, often forming a regulatory switch. To find evidence of this phenomenon in *M. tuberculosis*, we scanned consistently hypomethylated loci for nearby MTase motifs on the same strand (*Table 1*), including locus *cobK* 304. In most isolates, *cobK* 304 was only 8 bp distant from another HsdM site, *cobK* 312 (together making four palindromic motif matches within *cobK*). In these isolates, *cobK* 312 was methylated while cobK 304 was hypomethylated. However, in IO isolates the HsdM motif at *cobK* 312 was disrupted by a nearby deletion. If MTase crowding was responsible for the hypomethylation of *cobK* 304, then the removal of *cobK* 312 in IO isolates may be responsible for *cobK* 304

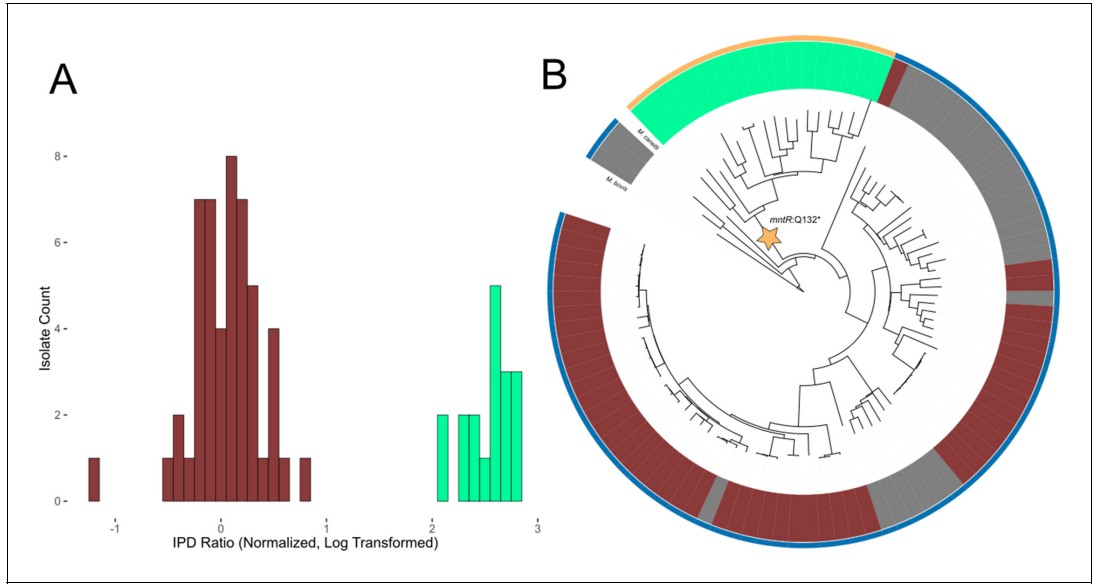

**Figure 5.** Evidence of transcription factor occlusion at hypomethylated MTase sites. (**A**) Histogram showing the distribution of IPD ratios at the HsdM motif locus cobK:304, 304 bp downstream from the start codon of gene cobK. Included isolates have active HsdM and possess the HsdM target motif at the cobK:304 locus. IPD ratios are normalized to the mean IPD ratio of adenine bases in their respective isolates (excluding bases targeted by known MTase motifs) and log$_2$-transformed. The histogram uses a bin width of 0.1. Red bars count isolates classified as 'hypomethylated' at the *cobK* site, while mint bars count isolates classified as methylated at the site. (**B**) Phylogenetic tree of the 90 clinical and reference *M. tuberculosis* isolates and 3 *M. africanum* isolates included in this study, along with outgroups *M. bovis* and *M. canetti*. Isolates are colored in the middle ring by their methylation status at the HsdM motif site *cobK*:304. Red isolates are classified as hypomethylated at the *cobK* site; green isolates are classified as methylated at the site, and gray isolates either have an inactive HsdM methyltransferase, or are missing the HsdM target motif 304 bp within their *cobK* gene. Isolates are colored in the outer ring by the genotype of their *mntR* (Rv2788) gene. *mntR* encodes for a transcription factor whose binding motif matches the context sequence of the *cobK* 304 site (p = 2.63×10$^{-5}$, converted log-likelihood ratio score). Gold isolates had the variant *mntR* Q131STOP, a nonsense mutation that introduces an early stop codon that truncated the gene and presumably removed its function. The blue isolates do not have a nonsense mutation, though one isolate had the missense mutation *mntR* P149L.

The online version of this article includes the following figure supplement(s) for figure 5:

**Figure supplement 1.** Frequency of hypomethylation calls in each *M. tuberculosis* clinical isolate.

**Figure supplement 2.** Distribution of IPD ratios in example *M. tuberculosis* clinical isolates, by Bayesian classification.

**Figure supplement 3.** Distribution of coverage values at MTase motif sites for indeterminate and determinate calls.

methylation in that lineage. As *cobK* 304 was consistent with both previously described phenomena, it is uncertain whether its hypomethylation was caused by its neighboring MTase motif, or by occlusion by transcription factor MntR.

## DNA adenine methylation is widespread and distinctly patterned at promoters

Next, we systematically probed promoters with MTase motif sites to identify common configurations between motif sites and characterized TSSs (*Cortes et al., 2013*; *Shell et al., 2015*). Within promoter regions (≤50 bp upstream from the TSS), targeted adenines of MamA and HsdM motifs had distinct peaks at the edges of the −10 element (*Figure 6A*). The MamA peak comprised 22 promoters coincident with the −10 element in the configuration that has been shown previously to modulate transcription (*Shell et al., 2013*) (4–5 and 7–8 bp upstream from TSS, *Figure 7*). These included the four shown to affect transcription (*Figure 7*, blue stars). Notably, none of these four were hypomethylated or hypervariable, indicating that lack of anomalous methylation in vitro does not preclude a role in transcriptional regulation. Common (n ≥ 75) HsdM motif sites overlapped with the −10 element of 33 promoters. While nineteen of these match those recently reported (*Chiner-Oms et al., 2019*), 13 are novel. These HsdM motif sites frequently overlap with the −10 promoter element in a configuration analogous to that common in MamA motifs, but on the distal (−10 to −13 bp) end (*Figure 8*). In total, 212 genes have common (in ≥75 isolates) promoter MTase motif sites (*Supplementary file 6*).

Next, we scanned for SFBS motifs overlapping promoter MTase motif sites. Sigma factors SigA and SigB overlapped MTase motif sites most frequently (*Figure 6B*), though SigC, SigD, SigI, and SigF overlapped motif sites as well (*Figure 6—figure supplement 1*, *Supplementary file 6*). MamB motif sites rarely overlapped SFBSs, while orphan MTase motif sites frequently did (*Figure 6A–C*). MamA motif sites were more frequent in promoter regions than HsdM sites, however HsdM sites more frequently overlapped a SFBS (*Figure 6B*, perhaps explainable by both HsdM and SigA target motifs including the dinucleotide 'TA'). The −10 promoter element guides formation of transcription initiation complexes (*Browning and Busby, 2016*), and DNA methylation alters biophysical properties that tune promoter strength (*Polaczek et al., 1998*) (DNA melting temperature *Gries et al., 2010* and DNA bending during open complex formation *Saecker et al., 2002*). This mechanism is particularly plausible for MTase motifs that overlap SigA SFBSs (*Figure 6B*). SigA is a sigma-70 homolog, which contacts the −10 promoter element at positions −7 through −12 upstream of the TSS (*Feklistov and Darst, 2011*), precisely where overlap with HsdM and MamA-methylated adenines are frequent (*Figure 6*). These findings provide a potential mechanism for cis-regulation of dozens of genes with orphan MTase motifs in *M. tuberculosis*.

Promoters of fifteen genes harbored hypervariable motif sites. Most (11/15) were hypervariable in both palindromic sites (*Figure 6D*). Seven motif sites comprise a cluster of hypervariable MamA motif sites in the spacer between the −10 and −35 promoter elements (19–24 bp range, *Figure 7*). While this region does not overlap with SFBSss, transcriptional effectors commonly bind here to tune gene expression (*Newberry and Brennan, 2004*; *Pandey et al., 2015*), providing a candidate mechanism driving the differences between strains and mechanistic plausibility for transcriptional influence at these sites. No MamB promoter motif sites were hypervariable (*Figure 6C*), consistent with a classic RM-system without regulatory roles, once again contrasting with the signatures of gene regulation present at orphan MTase sites.

## HsdM promoter methylation is associated with transcription levels of downstream genes

Notably, Rv1813c is hypervariable and has a motif site 11 bp upstream of its TSS, overlapping a SigA SFBS. Rv1813c was recently reported to be significantly under-expressed following ΔhsdM, but the authors did not identify the SigA overlap with this motif site in the Rv1813c promoter (*Chiner-Oms et al., 2019*). This discovery prompted us to re-evaluate the ΔhsdM differential expression results recently reported to have no direct influence on transcription at methylated promoters. In that work, the authors defined 'differentially expressed' genes using thresholds on both significance (adjusted p-value≤0.05) and magnitude (|log$_2$-foldchange| ≥ 1). Since we were interested in the mechanism (Does HsdM promoter methylation have the capacity to influence transcription?) rather

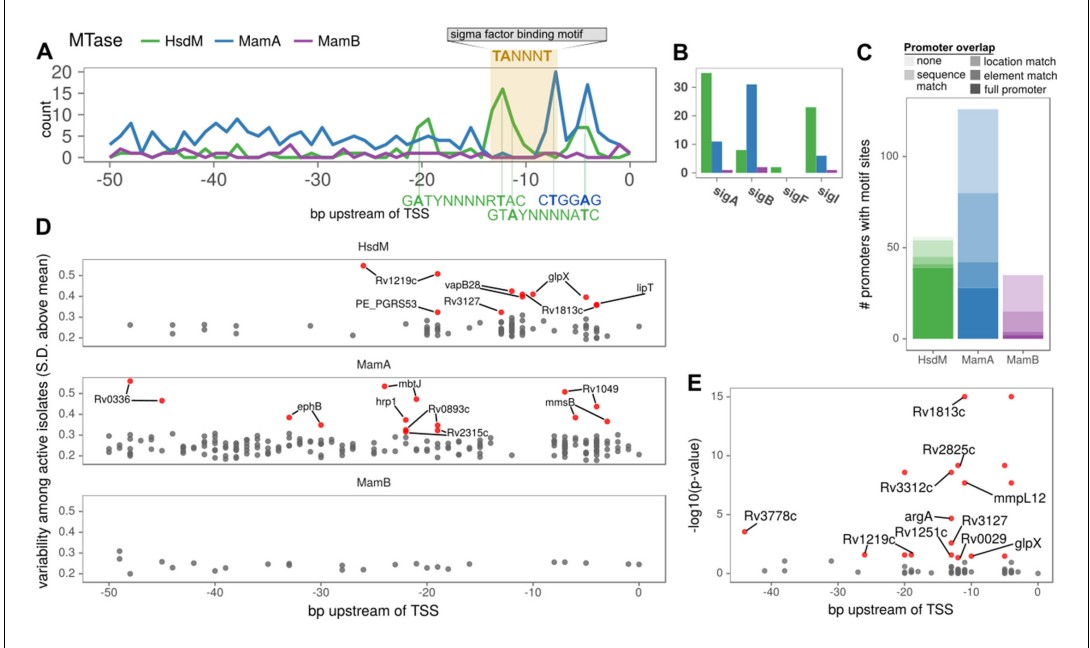

**Figure 6.** Configuration of orphan MTase motif sites at promoters suggests widespread epigenetic influence on transcription. (**A**) Consistent MTase-SFBS-promoter configuration. Number of promoters with MTase motif sites (in ≥ 50 isolates) by distance upstream of TSSs, for each MTase. The canonical SigA −10 element binding motif is superimposed for conceptual clarity, all motif sites within the −7 to −12 bp window upstream of annotated TSSs are included, irrespective of SFBS overlap. MTase Motifs for overrepresented configurations (peaks) are shown in the orientation and positions that explain the observed peaks. (**B**) The number of promoters with the −10 element overlapping an MTase motif site (≥30 isolates), for each MTase and sigma factor. (**C**) Stacked histograms of the number of genes harboring promoter motif sites for each MTase. Darker shades indicate overlap with progressively more substantiated overlap with promoter elements. In 'full promoters' MTase motifs overlapped a SFBS that is part of a classical promoter architecture (Materials and methods); 'Element' matches overlap either the −10 or −35 SFBS the expected distance from the TSS but have neither an extended −10 promoter element nor the complementary element; 'Location' matches are the distance upstream of the TSS expected to overlap with −10 or −35 elements but do not overlap known SFBS motifs; 'Sequence' matches coincide with SFBSs but not in the expected position with respect to TSS; 'none' are ≤ 50 bp upstream of a TSS but meet none of the aforementioned criteria. (**D**) Variability (SD of $\log_2$ (IPD Ratio) across isolates) across isolates with active MTase (y-axis) for common (≥75 isolates) promoter motifs and their distance upstream of the TSS (x-axis). Downstream genes of hypervariable motif sites (≥3 SD above the mean MamB variability, red). (**E**) Genes with annotated HsdM promoter motifs integrated with a recent ΔHsdM differential expression (DE) study (*Chiner-Oms et al., 2019*). All HsdM promoter motifs plotted by position within the promoter (x-axis) and Benjamini-Hochberg adjusted $-\log_{10}$ (p-value) for DE (y-axis). Downstream genes of motif sites of significantly DE genes (red, BH-adjusted p≤0.05) genes overlapping the −10 element (7 to 13 bp upstream of TSS) are labeled. The two genes without overlapping sites the −10 element have both their motif sites labeled (if within 50 bp).

The online version of this article includes the following source data and figure supplement(s) for figure 6:

**Source data 1.** Frequency and downstream CDS of frequent (in ≥ 30 isolates) MTase motif sites ≤ 100 bp upstream of a TSS.
**Source data 2.** MTase motif sites overlapping −10 sigma factor binding motifs 7–13 bp upstream of the TSS.
**Source data 3.** Promoters harboring MTase motifs and the extent of evidence for their overlap with promoter elements.
**Source data 4.** Re-analysis of differential transcription data following HsdM knockout.
**Source data 5.** Overlaps between −10 and −35 sigma factor binding sites and promoter MTase motif sites.
**Figure supplement 1.** Sigma factor binding site (SFBS) motif and MTase motif overlap.

than the magnitude of its effect, we defined differentially expressed genes according only to significance. With these criteria, 310 genes (*Figure 6—source data 4*, *Supplementary file 7*) were differentially expressed between $hsdM_{WT}$ and ΔhsdM (Δ*hsdM-DE*). Genes with HsdM motif sites in their promoters (n = 11) were significantly enriched among Δ*hsdM-DE* genes (p = 0.000215, OR = 4.47, 95% CI: 1.99–9.37, two-tailed Fisher's Exact; *Figure 6E*). Therefore, we conclude that HsdM promoter motifs are associated with expression change following HsdM knockout, though the magnitude of the effect is subtle in vitro ($|\log_2\text{-foldchange}| \leq 1.16$).

Nine of these 11 Δ*hsdM-DE* genes with HsdM promoter motifs overlapped with the −10 promoter element (*Figure 6E*), suggesting a direct effect on transcription analogous to the configuration for MamA previously described (*Shell et al., 2013*). Within these nine are 3 of the four genes

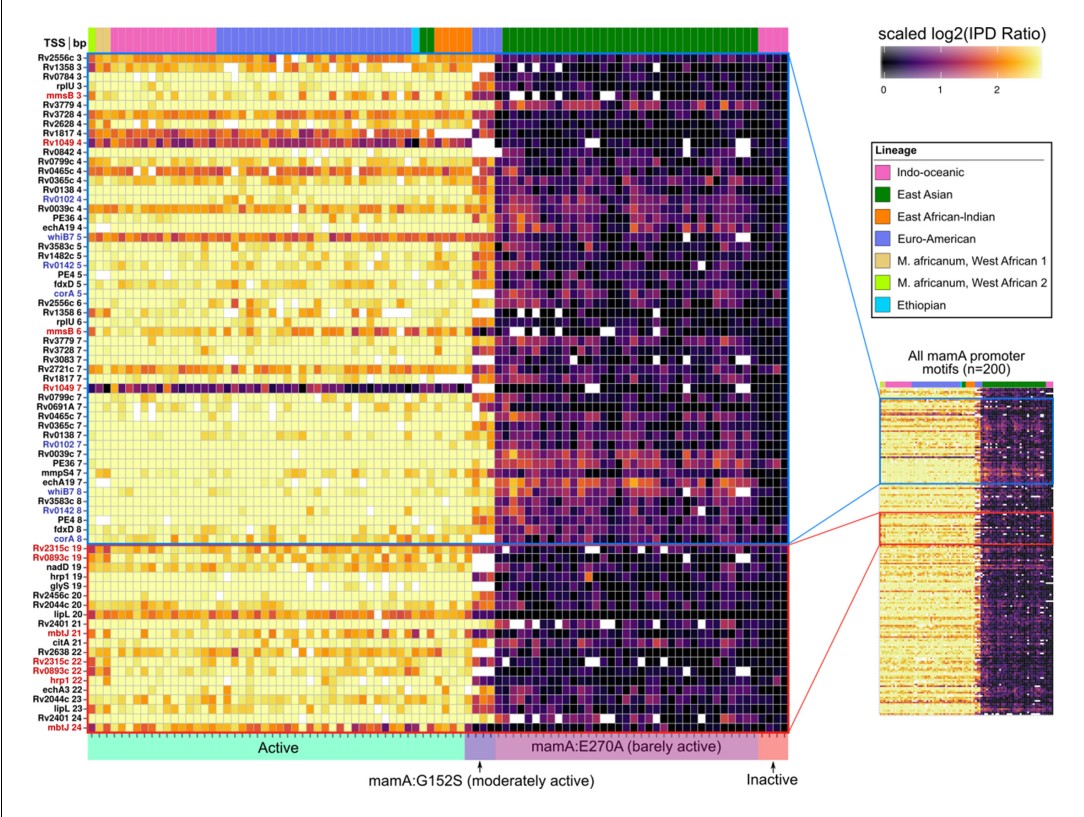

**Figure 7.** Methylomic variation at promoters harboring orphan MTase motifs: MamA motifs. Heatmap depicting degree of methylation (scaled $\log_2$ of IPD ratio averaged across reads) across all 93 clinical isolates (columns) at all common (present in $\geq$ 75 isolates) MamA promoter ($\leq$50 bp upstream of a TSS) motif sites (rows). The coloring scale of the heatmap boxes max out at the median scaled IPD across all motif sites across all isolates with active MamA allele and bottom out at 0 (corresponding to no methylation). For each isolate, HsdM activity (bottom) and lineage (top) are indicated. Isolates within each heatmap are sorted first by activity, and then by lineage. MamA motifs in configuration with −10 promoter element akin to that of the promoters shown to exhibit MamA-methylation-dependent transcriptional response under hypoxia (*Shell et al., 2013*) (blue pop-out) and those within a region with a high density of hypervariable sites (red pop-out) are highlighted. Color of axis labels highlights the specific motif sites shown by Shell and colleagues to affect transcriptional response to hypoxia (*Shell et al., 2013*) (blue) and motif sites hypervariable across isolates with active MamA (red).

The online version of this article includes the following source data and figure supplement(s) for figure 7:

**Source data 1.** Sequencing kinetics for all common MamA promoter motifs.

**Figure supplement 1.** Selected promoters with −10 SFBS-overlapping, hypervariable, or consistently hypomethylated motif sites.

with hypervariable HsdM-methylation in promoter motifs, suggesting that differential selection on the methylome during growth in vitro may affect gene expression.

## Discussion

Here, we assembled, annotated, and investigated DNA adenine methylomes of 93 *M. tuberculosis* complex (MTBC) clinical isolates from patients in high TB-burden countries (*Figure 1*), the largest survey of methylomic diversity in the MTBC to date. Through functional, comparative, integrative, and heterogeneity analyses of these methylomes, we identified drivers and sites of variability in DNA adenine methylomes across the MTBC. Mapping DNA methyltransferase (MTase) variants to methylomic data expanded the allele-function mappings for the three known MTBC DNA adenine methyltransferases (*Supplementary file 1*), clarified several disputed or errant reports of MTase function (*Chiner-Oms et al., 2019*; *Phelan et al., 2018*; *Shell et al., 2013*; *Figure 2—figure supplement 1*), and determined that the MTase variants $mamA_{E270A}$, $mamA_{G152S}$, and $mamB_{K1033T}$ conferred partial MTase activity (*Figure 2*). Heterogeneity analysis revealed that these three alleles drive intracellular stochastic methylation (*Figure 3A–G*), conferring Intercellular Mosaic Methylation (IMM) to 38/93

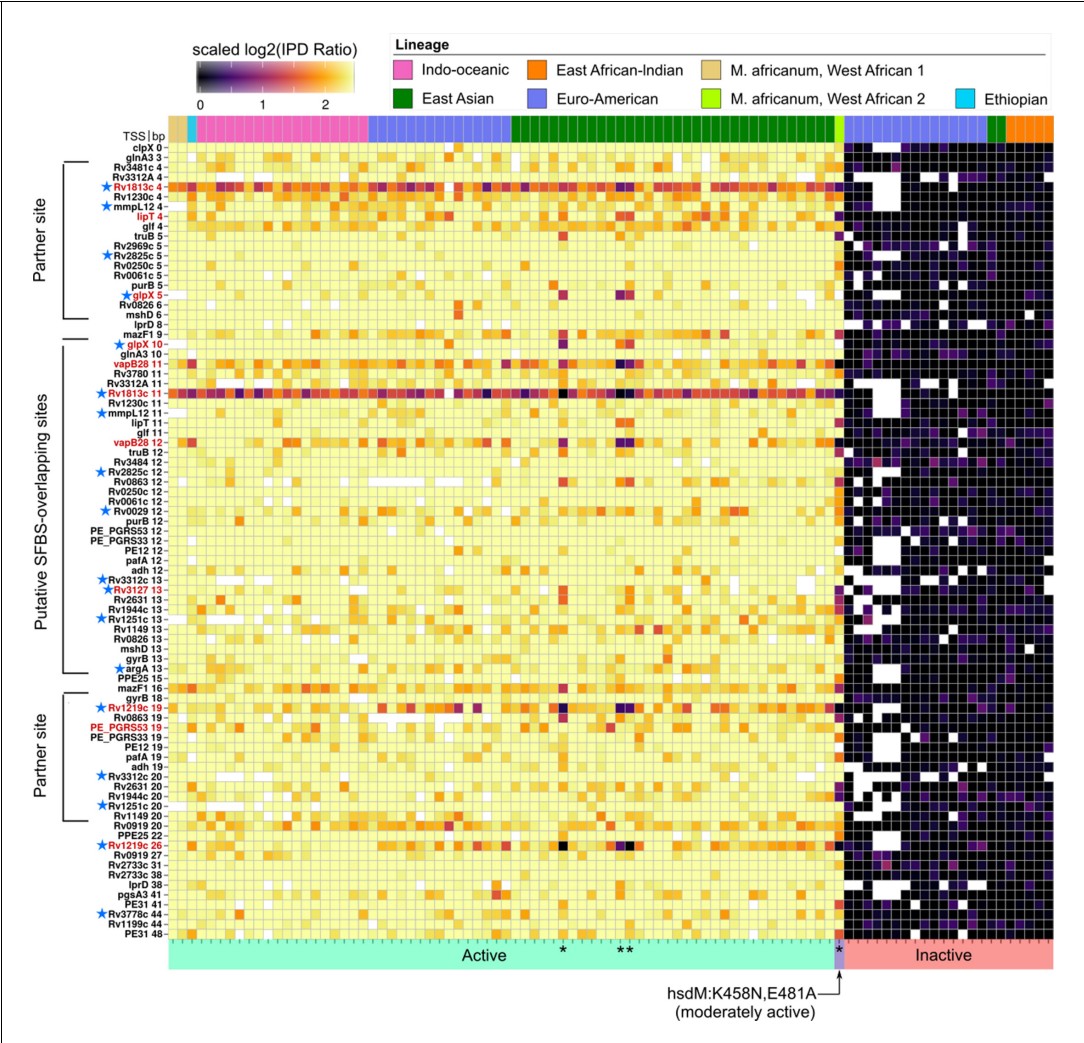

**Figure 8.** Methylomic variation at promoters harboring orphan MTase motifs: HsdM motifs. Similar to *Figure 7* but for HsdM motif sites. Heatmap depicting degree of methylation (scaled log$_2$ of IPD ratio averaged across reads) across all 93 clinical isolates (columns) at all common (present in ≥ 75 isolates) HsdM promoter (≤50 bp upstream of a TSS) motif sites (rows). The coloring scale of the heatmap boxes max out at the median scaled IPD across all motif sites across all isolates with an active HsdM allele and bottom out at 0 (corresponding to no methylation). For each isolate, HsdM activity (bottom) and lineage (top) are indicated. Isolates are sorted first by activity, and then by lineage. 'Putative SFBS-overlapping sites' are those with an analogous configuration with the −10 promoter element shown the MamA motif overlap highlighted in *Figure 7*, but overlapping the end of the −10 promoter element distal to the TSS, rather than the proximal end. 'Partner sites' are loci at the position that includes the palindromic partners of putative SFBS-overlapping sites. Promoter MTase motif sites with hypervariable kinetics across HsdM-active isolates (red text) or upstream of genes differentially expressed in HsdM knockout (blue stars) are indicated. Isolates with convergent methylation levels at a subset of notable loci despite having divergent HsdM genotypes and belonging to different lineages are indicated by asterisks (*).

The online version of this article includes the following source data for figure 8:

**Source data 1.** Sequencing kinetics for all common HsdM promoter motifs.

studied isolates, including 34/36 EAS isolates (*Supplementary file 2*). Comparative methylomic analysis identified subsets of motif sites consistently hypomethylated (*Table 1*, *Supplementary file 5*) and hypervariable across isolates (*Figure 4*, *Supplementary file 4*). Functional and integrative analyses uncovered previously unreported promoters harboring methylation motif sites (*Figures 6–8*, *Supplementary file 6*), implicating clinically important phenotypes as potentially regulated by DNA adenine methylation (*Figure 7—figure supplement 1*, *Supplementary file 6*), and put forth evidence that HsdM promoter methylation directly influences transcription methylation (*Figure 6E*), contradicting conclusions from previous work (*Chiner-Oms et al., 2019*). These findings add to the growing body of literature demonstrating bacterial epigenomics is an important complementary

focus to genetic and phenotypic analysis in studying microbial diversity, gene regulation, and evolution.

Sequencing kinetics of MTase target motif sites indicated heterogeneous methylation in isolates with MTase variants $mamA_{E270A}$, $mamA_{G152S}$, and $mamB_{K1033T}$ (*Figure 2A*). Read-level kinetic analysis confirmed this heterogeneity, and characterized the phenomenon as intracellular stochastic methylation, rather than phase-variable methylation (*Figure 3*). Further heterogeneity analysis demonstrated that methionine starvation can induce intracellular stochastic methylation in isolates with wild-type MTase activity (*Figure 3—figure supplement 1*). In stochastic methylation, the methylation status of each MTase target site varies independently between cells (*Beaulaurier et al., 2015*). The resulting subpopulations thus carry diverse combinations of methylated and unmethylated sites, a phenomenon we have termed 'intercellular mosaic methylation' (IMM, *Figures 3* and *9*). Thus, the potential diversity of DNA methylation patterns in IMM across cells scales logarithmically with the number of motif sites targeted by the MTase exhibiting IMM.

This is not the first report of mosaic-like patterning of DNA adenine methylation in prokaryotes. Mosaicism can result from independent ON/OFF switching of multiple phase-variable MTases (*Atack et al., 2018*) or from domain movement of the target recognition domain (TRD) (*Furuta and Kobayashi, 2012*), a phenomenon known as 'DoMo' (*Furuta et al., 2014*). However, IMM departs from these two previously described types of mosaic-like methylation heterogeneity in two important respects. First, in the degree of methylomic diversity it generates (*Figure 9A*). Just as independent state-changes of multiple modification enzymes (*Casadesús and Low, 2013*) increases the diversity of epigenetic bacterial lineages beyond that of individual phase variation systems, IMM extends this diversity further still, scaling logarithmically with the number of motif sites targeted by the stochastic MTase. In nature, the set of methylation states that manifest may be constrained below this theoretical set by a variety of mechanisms, such as interaction with DNA-binding proteins, or switch-like behavior between proximal MTase sites (*Casadesús and Low, 2013*). Nonetheless, the number of adoptable states is large enough that states are practically certain to differ between parent and daughter cells. Second, IMM is distinct in the pattern of epigenetic inheritance from parent

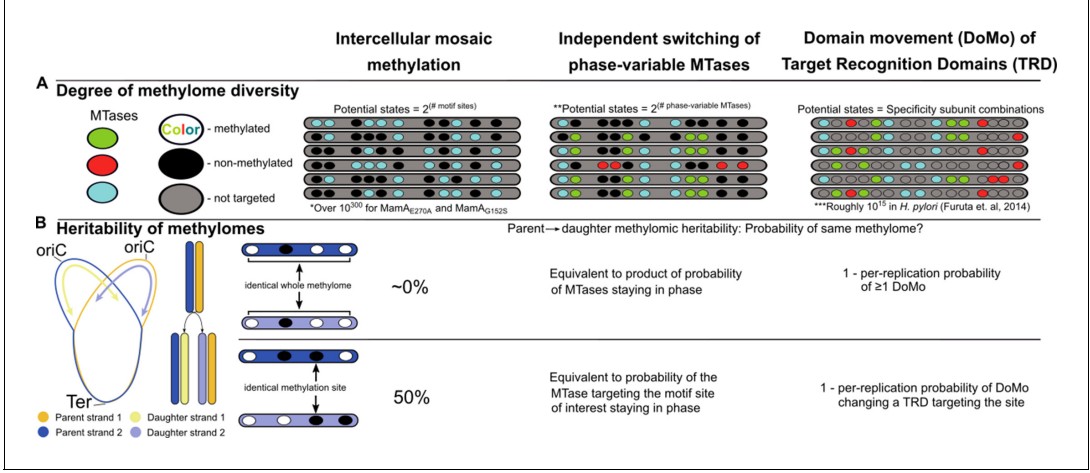

**Figure 9.** Intercellular mosaic methylation (IMM) is distinct from other forms of mosaic-like DNA methylation. Conceptual illustration contrasting DNA methylome diversification and epigenetic inheritance between IMM and other mosaic-like mechanisms of heterogeneous DNA adenine methylation. (**A**) Cartoon illustrating the nature of methylomic diversity depicts individual cells' chromosomes (gray bars) with methylation motifs (ovals). Oval colors represent distinct DNA methyltransferases (MTases). *Practically infinite, estimated as $2^{1,978}$ (there are roughly 1,978 MamA motif sites per replisome) under the assumption that methylation propensity on the daughter strand is independent from methylation status of other motif sites on the daughter strand and parent strand. **Assumes there are two phases. Some phase-variable MTases with more than two phases have been described. In these cases, potential states would be equivalent to the product of the sequence of numbers of phases for all independent phase-variable MTases. ***Calculated by Furuta and Kobayashi as the product of 1,000 DNA sequence specificities per MTase across 5 MTases in *Helicobacter pylori* (*Furuta and Kobayashi, 2012*). (**B**) Diagram illustrating the relationship between daughter and parent strains as it relates to conservation of the whole methylome (top) and at a single methylation site (bottom). Under the assumption of genuine stochasticity, IMM would practically never re-pattern the daughter strand identically to its parent. In contrast, the methylation status at any given methylation site would match between parent and daughter cells in 50% of cases.

to daughter cell (*Figure 9B*). In mosaic-like methylomes driven by independent switching of phase-variable MTases (canonically a frameshift) and DoMo (*Furuta et al., 2014*) (via homologous recombining of TRDs) the methylome-patterning determinant is passed genetically to the daughter strand, unless there is a phase change (*Sánchez-Romero and Casadesús, 2020*). IMM lacks a genetic basis for transgenerational methylome inheritance and no epigenetic mechanism of inheritance for IMM in *M. tuberculosis* is known at present. Consequently, our current knowledge suggests a greater degree of methylomic diversity is spread throughout the population in each replication event, but that any advantageous methylation patterns would lack a stabilizing mechanism. This apparent differential capacity to produce epigenetic lineages assumes that in IMM, the methylation status of parent strand motif sites has no effect on the probability of daughter strand motif sites being methylated. Empirically testing this assumption would refine our understanding of the implications IMM has for adaptive evolution.

Transcriptional influence by DNA adenine methylation is prevalent among human pathogens (*Casadesús and Low, 2013*). Previous study of *M. tuberculosis* genes with MamA-methylated promoters demonstrated that DNA adenine methylation directly influences the expression of genes with MamA target motifs in their promoters (*Shell et al., 2013*). Our promoter annotation expands the set of promoters potentially regulated by MamA-methylation (*Figure 7*, *Supplementary file 6*) and our Δ*hsdM*-DE analysis (*Figure 6E*) strongly suggests that HsdM-methylation at promoters directly influences transcription. This provides a potential mechanism in *M. tuberculosis* for heterogeneous methylation to produce heterogeneous phenotypes.

The physiological consequences of mycobacterial gene expression control by DNA methylation, however, remain to be identified. Therefore, the adaptive benefits of phenotypic heterogeneity ostensibly conferred by IMM remain hypothetical. Theoretical grounds for phenotypic heterogeneity to confer survival benefits have been described previously (*Wolf et al., 2005*), as have examples of adaptive phenotypic heterogeneity driven by DNA methylation in bacterial pathogens (*Atack et al., 2018*; *Low and Casadesús, 2008*). The adaptive benefit of any epigenetically influenced process depends on the degree to and manner in which it is heritable across generations. In the absence of an identified mechanism favoring self-perpetuity, our description of IMM provides bacterial populations with a means for retaining adaptive methylation patterns, but not for amplifying them. Without heritability, IMM-driven phenotypic heterogeneity could still confer adaptive benefit through a 'bet-hedging' strategy (*Casadesús and Low, 2013*; *Wolf et al., 2005*). This hypothesis is consistent with observations of *M. tuberculosis* 'persister cells' (*Vilchèze et al., 2017*), minority groups that are pre-adapted to tolerate initial exposure to macrophage (*Daniel et al., 2011*) and drug pressure (*Keren et al., 2011*), by entering dormancy (*Jain et al., 2016*) or activating efflux pumps (*Colangeli et al., 2005*; *Mustyala et al., 2016*).

Future work examining changes in methylation patterns in strains with IMM-conferring MTases under selective pressure would help determine whether DNA adenine methylation patterns are heritable in *M. tuberculosis*, and whether IMM is compatible with the hypothesis of 'epigenetic-driven adaptive evolution' (*Furuta and Kobayashi, 2012*). The hypervariability of numerous orphan MTase loci (n = 344, *Supplementary file 4*) could be taken to imply differential selection of methylation patterns across isolates (in vitro). Two observations provide circumstantial evidence in support of this interpretation. First, hypomethylation (which coincides with many hypervariable sites, *Table 1*, *Supplementary files 4* and *5*) is a signature of epigenetic regulation in prokaryotes (*Blow et al., 2016*), suggesting genes with hypervariable promoter motif sites (n = 11, *Figures 6–8*) are epigenetically regulated, providing a potential basis for differential selection across strains. Second, several sites are methylated at similar levels between three EAS isolates and a genetically distant (SNP distance ≥ 2,722) *M. africanum* isolate (*Figures 4A* and *8*) suggesting convergent epigenomic selection in vitro. This convergence was not driven by MTase genotype, as the isolates had discordant HsdM alleles. Notably, convergently methylated sites between these isolates included Δ*hsdM*-DE persistence (*raaS*) and dormancy (*Rv1813c* and *glpX*) genes (*Figure 6*), suggesting methylation at these sites affect promoter strength and raising the possibility that these methylation patterns have important phenotypic consequences. However, both these observations may be due to a secondary effect, such as MTase antagonism driven by mutations in DNA-binding proteins, as we observed at *cobK* 304 (*Figure 5*), or some other primary factor. Taken together, the evidence from this work is insufficient to conclude that any sites are under epigenetic selection.

Our promoter methylation analysis produced different results than a recently published analysis (*Chiner-Oms et al., 2019*) and reached opposing conclusions for the role of HsdM in regulating promoter strength. While the authors conclude that 'methylation seems to play a minimal role in shaping in-vitro gene expression', integration of our HsdM motif-harboring promoters with their Δ*hsdM* RNAseq data shows a clear association between HsdM promoter methylation and in vitro gene expression (*Figure 6E*). We believe differences in approach drove our disparate conclusions. For their analysis, Chiner-Oms and colleagues relied on reference-mapping and a single source of TSSs, and they focused on SigA motifs to identify promoter MTase motifs (*Chiner-Oms et al., 2019*). In contrast, we transferred TSS annotations from two *M. tuberculosis* transcriptomic studies (*Cortes et al., 2013*; *Shell et al., 2015*) to contiguous regions of finished de novo assemblies, and scanned for SigA-SigM SFBS motifs (*Chauhan et al., 2016*) and promoters lacking known SFBSs. These differences explain why our analysis captured the association between HsdM-methylation and expression of downstream genes.

By integrating DNA adenine methylomes with fully-annotated genome assemblies and TSS-mapping data, this work generates a corpus of putative interactions between DNA methylation and regulatory effectors (*Supplementary files 4–7*), providing a basis for generating specific, testable functional hypotheses for DNA adenine methylation in *M. tuberculosis*. The functions of genes with promoter motif sites that were hypervariable (*Figures 7* and *8*, *Figure 7—figure supplement 1*) or in stereotyped configurations with −10 promoter elements (*Figures 6* and *8*; *Supplementary file 6*) implicate clinically important processes as under control of DNA adenine methylation (*Supplementary files 1* and *6*, *Figure 7—figure supplement 1*). These include drug resistance, metal ion homeostasis (*corA* (*Park et al., 2019*), *lpqS* (*Darwin, 2015*), *mmpS4* (*Jones et al., 2014*), *higA* (*Schuessler et al., 2013*), *mbtJ* (*Chownk et al., 2018*), and *hemN McMahon et al., 2012*), and the induction and maintenance of dormancy—all functions with examples of modulation by DNA adenine methylation in other human pathogens (Beaulaurier et al.; *Brunet et al., 2020*; *Cohen et al., 2016*; *Sánchez-Romero and Casadesús, 2020*). The resistance-implicating genes among these mediate resistance through gene regulation (*whiB7*-controlled expression of *eis*, *tap*, and *Rv1473* (*Duan et al., 2019*; *Morris et al., 2005*), and *raaS*-controlled expression of *Rv1218c* and *Rv1217c Wang et al., 2013*), drug efflux (*drrA* (*Mustyala et al., 2016*), *iniA* (*Colangeli et al., 2005*), *Rv3728* (*Gupta et al., 2010*), and efflux-targets of *whiB7* and *raaS*), and other mechanisms (*glf* (*Chen and Bishai, 1998*), *mshC* (*Parida et al., 2015*), *mshD* (*Vilchèze et al., 2008*), *pafA* (*Samanovic and Darwin, 2016*), *Rv3050c* (*Nieto R et al., 2018*), and *gyrB* (*Nosova et al., 2013*).

Our finding that most (34/36, *Supplementary file 2*) Beijing clinical isolates exhibited constitutive IMM prompts the question of whether IMM might contribute to their global success. Methylated promoters implicate some hallmarks of the Beijing sublineage: facile dormancy induction, increased host-lipid utilization, TAG accumulation in aerobic environments (*Reed et al., 2007*), and increased synthesis of cell envelope components and virulence lipids (*Huet et al., 2009*; *Figure 7—figure supplement 1*, *Supplementary file 6*). While some of these hallmarks have been attributed to genetic factors, such as higher basal expression of the DosR-regulon (*Reed et al., 2007*), gaps remain in our understanding. One hypothesis is that, in Beijing isolates, MamA$_{E270A}$-driven mosaicism confers phenotypic heterogeneity, thereby enabling access to a greater number of phenotypic solutions during adaptive evolution. This hypothesis could be investigated by exposing MamA$_{WT}$ and MamA$_{E270A}$ mutants to different selection pressures and comparing evolutionary outcomes and methylomes of the adapted strains.

We cannot extrapolate directly from the DNA methylation patterns we report here to what occurs during infection. Sequencing kinetics are measured from DNA extracted after extensive culturing, during which any methylomic adaptation to the host environment may have changed. Directly sequencing from patient specimens would be ideal to assay DNA methylation patterns in vivo, but DNA input requirements necessitate culturing prior to DNA extraction (*PACBIO, 2013*), as DNA amplification erases epigenetic markings. Studying methylomic adaptation to host-like conditions (e.g. hypoxia, host-lipids as carbon source) can reveal context-dependent selection of methylation patterns, and time-course serial sequencing could characterize the dynamics of their selection. Coupling these sequencing studies with transcriptomic, proteomic, and phenotypic assays could clarify the relationship between DNA adenine methylation, gene expression, and phenotype.

Throughout this manuscript we have referred to the DNA adenine methyltransferase encoded by Rv2756c as HsdM (*hsdM* for the gene) and its specificity subunit encoded by Rv2761, as HsdS (*hsdS* for the gene) to be consistent with previous work (*Shell et al., 2013*). It appears that Rv2756c was originally referred to as HsdM based on homology to *hsdM* in R-M systems—before the existence of its restriction component had been investigated—and has propagated through subsequent studies (*Chiner-Oms et al., 2019*; *Gomez-Gonzalez et al., 2019*; *Phelan et al., 2018*; *Zhu et al., 2016*). However, it has since been determined that Rv2756c lacks a functional HsdR component (*Zhu et al., 2016*). According to the prevailing nomenclature conventions, the symbol 'hsd' is for Type 1 R-M systems (*Loenen et al., 2014*; *Roberts et al., 2003*), which Rv2756c is not part of, since it lacks a functional restriction component. Therefore, we propose that the orphan methyltransferase encoded by Rv2756c be renamed to MamC (*mamC* for the gene) <u>M</u>ycobacterial <u>A</u>denine <u>M</u>ethyltransferase <u>C</u> (since MamA and MamB are assigned to other mycobacterial DNA adenine methyltransferases). Likewise, we propose that the specificity subunit of MamC encoded by Rv2761 be renamed to *mamS*/MamS (formerly *hsdS*/HsdS) and the specificity subunit fragment encoded by Rv2755 (formerly *hsdS.1*/HsdS.1) to *mamS.1*/MamS.1. This proposed nomenclature retains the S and S.1 from *hsd<u>S</u>* and *hsdS.1*, is consistent with the extant naming convention of MamA and MamB, and removes the erroneous implication that HsdM/HsdS/HsdS.1 are part of a Type 1 R-M system.

This integrative analysis of 93 clinical isolates' DNA adenine methylomes spotlights DNA methylation as a fundamental source of diversity in the MTBC. The results provide a basis for further investigation of the roles played by DNA adenine methylation in *M. tuberculosis'* physiology and adaptive evolution. In particular, the discovery of constitutive IMM-driven by MTase genotype raises the possibility that DNA adenine methylation translates into differences in adaptive capacity between MTBC strains.

## Materials and methods

### Isolate acquisition and inclusion criteria

MTBC colonies were isolated from sputa of 154 tuberculosis patients, originating from Hinduja National Hospital (PDHNH) in Mumbai, India; Phthisiopneumology Institute (PPI) in Chisinau, Moldova; Tropical Disease Foundation (TDF) in Manila, the Philippines; The National Health Laboratory Service of South Africa (NHLS) in Johannesburg, South Africa; and World Health Organization Supranational References Laboratories in Stockholm, Sweden and Antwerp, Belgium (PRJNA555636). Of these 154 isolates, 113 were originally collected by the Global Consortium for Drug-resistant Tuberculosis Diagnostics (*Hillery et al., 2014*) and chosen for resequencing.

We also included two technical replicate control runs of avirulent reference strain H37Ra reported in a previous paper (PRJNA329548) (*Elghraoui et al., 2017*). An additional 24 publicly available SMRT-sequencing reads of clinical *M. tuberculosis* and *M. africanum* isolates were downloaded from the Sequence Read Archive, along with a triplicate run of virulent type strain H37Rv, and triplicate samples of a *metA* knockout strain of H37Rv before and after five days of methionine starvation (PRJEB8783) (*Berney et al., 2015*).

Of these isolates, 97 clinical *M. tuberculosis* isolates, 3 *M. africanum* isolates, and nine reference strains passed assembly quality control. Of them, seven clinical isolates and one *metA* knockout isolate failed our methylome pipeline (five isolates had multiple contigs and three isolates had position inconsistencies between their kinetics data and consensus sequence FASTA file). In total, our downstream analysis included 93 MTBC clinical isolates (including 3 *M. africanum*) and 8 runs of reference strains and laboratory mutants.

### Sample preparation and extraction

The *M. tuberculosis* and *M. africanum* samples were prepared and extracted at the World Health Organization Supranational Reference Laboratory in Stockholm, Sweden, and the Institute for Genomic Medicine at the University of California, San Diego in La Jolla, CA, USA. DNA preparation and extraction was performed as previously described (*Elghraoui et al., 2017*). All samples were streaked for isolation using standard microbiological methods, after which well-separated colonies were selected, emulsified, and sub-cultured on Löwenstein–Jensen slants and incubated until growth of a full bacterial lawn. To ensure enough bacterial material for extraction of sufficient high molecular

weight DNA for amplification-free SMRT-sequencing well-outgrown solid cultures (Löwenstein–Jensen medium). Typically, enough bacterial cells were obtained after 3–4 weeks of culture. At this stage, cultures reached the early- to mid-stationary phase. DNA was extracted from cultures in the stationary growth phase. DNA was extracted using Genomic-tips (Qiagen Inc, Germantown, MD, USA) following the manufacturer's sample preparation and lysis protocol for bacteria with the following modifications. Each culture was harvested directly into buffer B1/RNAse solution, homogenized by vigorous vortex mixing and inactivated at 80°C for 1 hr. Lysozyme was added and incubated at 37°C for 30 min followed by the addition of proteinase K and further incubation at 37°C for an additional 60 min. Buffer B2 was added and the mixture was incubated overnight at 50°C. The remainder of the Genomic-tip protocol was carried out exactly as described by the manufacturer. DNA purity and concentration were analyzed on a Nanodrop 1000 (Thermo Scientific, Waltham, MA, USA).

## DNA sequencing

DNA sequencing was performed at the Institute for Genomic Medicine at the University of California, San Diego. DNA libraries for PacBio (Pacific Biosciences, Menlo Park, CA) were prepared using PacBio's DNA Template Prep Kit with no follow-up PCR amplification. Briefly, sheared DNA was end repaired, and hairpin adapters were ligated using T4 DNA ligase. Incompletely formed SMRTbell templates were degraded with a combination of Exonuclease III and Exonuclease VII. The resulting DNA templates were purified using SPRI magnetic beads (AMPure, Agencourt Bioscience, Beverly, MA) and annealed to a twofold molar excess of a sequencing primer that specifically bound to the single-stranded loop region of the hairpin adapters. SMRTbell templates were subjected to standard SMRT-sequencing using an engineered phi29 DNA polymerase on the PacBio RSII system according to manufacturer's protocol.

## Genome assembly

For isolates that were sequenced on multiple SMRT cells, all SMRT cell raw reads were combined and assembled with either HGAP2 (*Chin et al., 2013*) or canu (*Koren et al., 2017*) with default parameters. Circularization was then performed to confirm a circular genome using minimus2 from amos or circlator (*Hunt et al., 2015*). Gene *dnaA* was set as the first gene in each genome. Iterative rounds of consensus polishing using BLASR (*Chaisson and Tesler, 2012*) and Quiver were executed three times. Default parameters were used except max coverage was set to 1000 for Quiver. Genomes failed assembly quality control if they could not be circularized, if their consensus polishing resulted in five or more variants after three iterations, or if PBHoney (*English et al., 2014*) detected a structural variant in the assembly supported by at least 10% of the reads. PBHoney was run with default parameters. Full details of methods are described by Ramirez-Busby and colleagues (manuscript in preparation).

## Lineage determination

For the isolates originally collected by the Global Consortium for Drug-resistant Tuberculosis Diagnostics, lineage information was obtained by inputting the MIRU-VNTR and spoligotype patterns determined previously[3] into TBInsight (*Shabbeer et al., 2012*). For all other genomes, a custom script, MiruHero (https://gitlab.com/LPCDRP/miru-hero), determined lineage. MiruHero takes in FASTA files with the whole genome sequences *M. tuberculosis* strains and in silico determines the strain's spoligotype and MIRU type. MiruHero then determines the strain's lineage by applying the same spoligotype and MIRU type interpretation rules used by TBInsight.

## Genome annotation

Gene annotations were transferred to each isolate from a well-characterized reference, virulent *M. tuberculosis* type strain H37Rv (*Lew et al., 2011*) with additional functional annotations curated from literature (*Modlin et al., 2018*). The transfer step was implemented using the Rapid Annotation Transfer Tool (RATT *Otto et al., 2011*) with the 'Strain' parameter. For each isolate, RATT read a FASTA file with the isolate's whole genome sequence, and read the curated reference annotation EMBL file of H37Rv, then created an EMBL annotation file for the isolate. RATT transfers annotated genome features to an isolate in regions of sequence similarity to the reference, adjusting genome position based on synteny blocks. RATT transferred both genes and transcriptional start sites (TSS)

from our curated H37Rv annotation. These TSSs were originally determined experimentally in the H37Rv strain by *Cortes et al., 2013* and *Shell et al., 2015*.

## MTase genotyping

To determine the genotype of the MTase genes *mamA* (*Rv3263*), *mamB* (*Rv2024c*), and *hsdM* (*Rv2756c*)/*hsdS* (*Rv2761c*) in each isolate, first eggNOG-mapper (*Powell et al., 2012*) identified these genes in each clinical isolate. However, MamB and HsdM are inactive in virulent type strain H37Rv (*Zhu et al., 2016*), we did not use the H37Rv genes as the wild-type allele. Instead, sequencing kinetics and the previously characterized target motifs were used to determine which isolates had active copies of each MTase gene, and the most common allele among active isolates was defined as the wild-type sequence. To call variants in these genes using these wild-type sequences, BLASTn then aligned the wild-type sequences against all genes predicted in each isolate by Prodigal (*Hyatt et al., 2010*). Each matching nucleotide sequence was translated into an amino acid sequence using transeq (EMBOSS 6.6.0.0, available online at www.ebi.ac.uk/Tools/emboss/transeq/index.html) to obtain nonsynonymous variants and truncations. The amino acid sequences were then aligned using MAFFT (*Katoh and Standley, 2013*) v7.205 with the – clustalout option, and a custom script converted the alignment to a genotype. To verify the identity of the 1356 bp *mamB* insertion variant, BLASTn aligned the *M. tuberculosis* insertion element IS6110 against the variant *mamB* gene (*Figure 2—figure supplement 1*). The IS6110 sequence is included in *Supplementary file 8*.

## Phylogenetic analysis

The genomes of *Mycobacterium canettii* CIPT 140070010 (NC_019951.1) and *Mycobacterium bovis* BCG Pasteur 1173P2 (AM408590.1) were used as outgroups. Then, dnadiff (*Marçais et al., 2018*) (v1.3) aligned each assembled genome to *M. tuberculosis* H37Rv (NC_000962.3) to call SNPs and small indels with default parameters. A custom Perl script then converted the out.snps file from dnadiff for each isolate into a VCF v4.0 file. A multi-FASTA file of concatenated variants was then created using each isolate's VCF file. The resulting multi-FASTA file was used to create a maximum likelihood phylogenetic tree using RAxML version 8.2 (*Stamatakis, 2014*), specifying a general time-reversible model of nucleotide evolution with 100 bootstrap replicates. All other settings were default. To visualize the phylogenetic distribution of MTase genotypes, a custom python script converted a CSV file with the MTase genotypes of each isolate into a tree color annotation file. The tree color annotation file and RAxML tree file were then uploaded to the Interactive Tree of Life (iTOL) (*Letunic and Bork, 2016*) webtool for visualization.

## Determination of IPD of MTase motif sites

To determine the IPD ratio at each nucleotide in each isolate, we ran SMRTanalysis with the Base Modification Detection with Motif Finding protocol with default parameters. A custom R script then scanned the FASTA sequence file of each isolate for matches to the MTase target motifs previously characterized in *M. tuberculosis* (*Zhu et al., 2016*) (https://gitlab.com/LPCDRP/dna-methylation/-/tree/publication/targets), then extracted the IPD ratio of the targeted adenine in each matching site from the Base Modification output. These IPD ratios were then log-transformed to produce a normal distribution and standardized by subtracting the mean IPD ratio (also log-transformed) of all adenines outside of MTase motifs in the isolate.

## MTase motif locus assignment

To track MTase motif sites across isolates, each MTase motif site in each isolate was assigned a locus tag based on the nearest gene. For each isolate, a custom python script read a CSV file with the genome positions of each MTase motif site in the isolate, and the isolate's genome annotation EMBL file. Using the genome positions of annotated coding sequences (CDS), the script identified the nearest gene boundary (CDS start or CDS stop) to each MTase motif site and assigned each site a locus tag. The locus tag contained the gene name of the neighboring gene boundary, the distance in nucleotides between the MTase site and the gene's CDS start position, an indicator whether the MTase site was on the same strand as the gene (sense) or the opposite strand (antisense), and an indicator whether the gene was downstream or upstream of the CDS start.

## Characterizing the kinetic error profile across technical replicates

We characterized the error profile of sequencing kinetics by comparing the IPD ratio of each base between replicate sequencing runs, and measuring how that variance increased with coverage. We used two sequencing runs on DNA isolated from avirulent type strain H37Ra sequenced at UCSD, and two publicly available sequencing runs on virulent type strain H37Rv (Bioproject accession PRJEB8783). We first log-transformed the IPD ratio of each base to account for the skewed distribution of ratio values (*Figure 1—figure supplement 1A–B*). We then plotted the difference in log-transformed IPD ratio of each base between runs by genome position (*Figure 1—figure supplement 1C*) and by per base coverage (*Figure 1—figure supplement 1D*). Using R's cor.test function with the 'pearson' method, we tested the significance of the correlation between per base coverage and the difference in log-transformed IPD ratio of each base between runs (*Figure 1—figure supplement 1D*).

## Characterization of MTase activity of MTase genotypes

For each MTase and each isolate, a custom R script plotted the distribution of log-transformed, scaled IPD ratios of each MTase motif site (*Figure 2A & D*). For all three MTases, MTase genotype reliably corresponded to distribution of sequencing at their respective motif sites. For most isolates, sequencing kinetics centered around 0 (consistent with no methylation) or around a narrow band consistent with full m6A methylation. MTase genotypes for these cases were labeled accordingly as 'fully active' or 'inactive' (loss-of-function). MTase genotypes for the remaining minority of cases, where sequencing kinetics distributed around a mean between 0 and the value around which fully active isolates distributed, were labeled as 'partially active'.

## Identification of hypervariable MTase motif loci

For each MTase, a custom R script found the set of MTase motif site loci present in at least 75 isolates. For each locus, summary statistics (mean, median, and standard deviation) of mean $\log_2$ (IPD Ratio) were calculated exclusively from isolates with active genotypes of the relevant MTase. The same was then performed to obtain median and standard deviation of mean $\log_2$ (IPD Ratio) for inactive isolates of each activity profile for each MTase, and, for MamA, for isolates with the common E270A *mamA* genotype. Hypervariable HsdM, MamA, and MamB motif sites were classified as those more than 3 s.D above the mean variability (standard deviation across isolates with active MamB at that site) for MamB motif sites, since they had the fewest outliers and are not an orphan MTase.

## Heterogeneous methylation analysis

SMALR (*Beaulaurier et al., 2015*) requires a de novo assembled genome FASTA file, a target motif, and a cmp.h5 file with aligned reads, to extract the IPD data from each MTase target motif site within each read. We ran SMALR on 96 samples using the MamA target motif CTGGAG. The isolates had the following *mamA* genotypes: 49 were wild-type, four were W136R, four were G152S, and 34 were E270A. We also ran SMALR on 70 isolates using the MamB target motif CACGCAG. The isolates had the following *mamB* genotypes: 60 were wild-type, nine had an loss-of-function variant, and one had the partially active variant K1033T. We then filtered out isolates with fewer than 20 total reads from downstream analysis. We additionally ran SMALR with the MamA target motif CTGGAG on one isolate assembled from published sequencing reads from a ΔmetA mutant of H37Rv that was SMRT-sequenced following 5 days of methionine starvation (*Berney et al., 2015*).

For each isolate, a cmp.h5 was generated by aligning its reads to its assembled FASTA file using BLASR (*Chaisson and Tesler, 2012*). SMALR was run on each isolate with the SMp (single-molecule, pooled distribution) argument. For MamA sites, the motif argument was set to CTGGAG, the modified position within the motif to 5, and the minimum number of motif sites per read to 6. For MamB sites, the motif argument was set to CACGCAG, the modified position to 6, and the motifs per read threshold to 6. The native IPD value of each read was used in place of SMp score. This substitution is susceptible to noise from local sequence contexts, but should still resolve differences between isolates and distinguish methylated and unmethylated reads (*Beaulaurier et al., 2015*). The distribution of native IPD values within each isolate for MamA and MamB sites was visualized using ggplot2 (*Wickham, 2016*), and comparisons between genotypes was performed using two-tailed Student's

t-tests with the t.test and effect sizes estimated with cohen.d functions in R (*R Development Core Team, 2018*).

## Identification of promoters

To identify MTase motif sites in probable gene promoters, for each isolate a custom python script read a CSV file with the genome positions of each MTase motif site in the isolate, and the isolate's genome annotation EMBL file. The script scanned the surrounding sequence on either strand of each MTase motif site for Sigma Factor Binding Sight (SFBS) motifs previously characterized in *M. tuberculosis* (*Chauhan et al., 2016*) (https://gitlab.com/LPCDRP/dna-methylation/-/tree/publication/targets). If a SFBS motif match overlapped with an MTase motif site, the MTase motif site was labeled with the sigma factor type corresponding to the SFBS motif (Sigma Factor A through Sigma Factor M, −10 element or −35 element). The script then compared the genome position and strand of the SFBS motif match to the genome position and strand of each annotated TSS in the isolate's genome annotation EMBL file. To meet the most stringent criteria for probable promoter sites ('full promoters', *Figure 6C*) an SFBS motif had to be the appropriate nucleotide distance upstream from an annotated TSS. If the SFBS motif match was the −10 component of a SFBS motif, the script checked if a there was a TSS on the same strand with a genome position 8 to 12 bp downstream of the matching sequence. If the SFBS match was a −35 component of a SFBS, the script instead checked for a TSS between 30 and 40 bp downstream. To capture MTase promoter interactions excluded by this conservative definition of SFBS, for the analysis in *Figure 6C*, categories were assigned indicating the substantiveness of the evidence supporting overlap with promoter elements. The following logic was executed programmatically in R to assign categories: 'Element' matches overlap either the −10 or −35 SFBS the expected distance from the TSS (see above) but have neither an extended −10 promoter element nor the complementary element; 'Location' matches are at the appropriate distance upstream of the TSS to overlap with −10 or −35 elements but do not overlap known SFBS motifs; 'Sequence' matches coincide with SFBSs but not in the expected position with respect to TSS; 'none' are within 50 bp upstream of a TSS but meet criteria for none of the aforementioned categories.

## Bayesian classification of base-specific methylation status

Even within isolates with active MTase genotypes, not every base with an MTase target motif was methylated. To identify MTase motif sites with no base modification (hypomethylated sites) despite an active MTase we took a Bayesian approach. In each isolate a custom R script estimated the distribution of normalized IPD ratios among unmodified bases by calculating the standard deviation and mean normalized IPD ratios of bases not within MTase motifs. The script then estimated the distribution of methylated bases by calculating the standard deviation and mean of bases targeted by MTase motifs. This estimate assumed that most bases targeted by MTase motifs were methylated, which held true in isolates with active MTase genotypes (*Figure 2A*). For each MTase motif site, the script calculated the conditional probability of the base belonging to either the modified or unmodified population, given its normalized IPD ratio and coverage. The script classified all bases more than nine times more likely to belong to the unmodified population as hypomethylated, all bases more than nine times more likely to belong to the modified population as methylated, and the remaining bases as indeterminate.

The coverage of each MTase site in each isolate was used to adjust the standard deviation of the distributions used to calculate its conditional probability, as bases with lower coverage have less consistent IPD ratios (*Figure 1—figure supplement 1D*). To perform this coverage adjustment, for each isolate we trained a model to estimate the expected standard deviation of any base given its coverage. After log-transforming and normalizing the IPD ratios of all bases in an isolate, the script calculated each base's number of standard deviations from the median normalized IPD ratio. Next, linear regression estimated the relationship between these standard deviations and the inverse coverage of each base. The resulting model estimated the standard deviation for each possible coverage value. When estimating the conditional probabilities of each MTase motif site, the code first calculated the mean and standard deviation of normalized IPD ratios in adenines within and without MTase motifs. It then multiplied these two standard deviations by the standard deviation predicted from the sequencing coverage at that MTase motif site. These adjusted standard deviations were

then used to estimate the distribution of normalized IPD ratios and calculate the conditional probability of the MTase motif site belonging to those distributions.

## Conserved hypomethylation patterns

Using the Bayesian classification of each MTase motif target and the loci labeled by our methylome annotation pipeline, we searched for hypomethylated loci that occurred in multiple isolates. For each locus, a custom R script counted the number of isolates with that locus, including only isolates with active genotypes of the MTase targeting the locus. Our script also counted the number of these isolates in which the locus was hypomethylated. To estimate the significance of these findings, we used a cumulative binomial test, with the first count as the sample size and the second count as the number of successes. To find the probability of hypomethylation for each Bernoulli trial if hypomethylation occurred randomly, we calculated the total frequency of hypomethylation among MTase motif sites in active isolates. A separate per trial probability was calculated for MamA, MamB, and HsdM. The Bonferroni correction adjusted for multiple hypothesis testing, by dividing the significance threshold by the total number of unique loci in this study. No reference strains were included in the analysis of hypomethylated loci.

## Transcription factor binding motif scanning

We searched for Transcription Factor (TF) binding motifs near hypomethylated bases using the command line motif scanner FIMO (*Grant et al., 2011*) version 4.12.0. For each hypomethylated locus in an MTase target motif, we extracted the sequence surrounding the locus, 20 bases on each side in a randomly selected representative isolate (only isolates hypomethylated at that locus were chosen). The context sequences were combined into a multisequence FASTA file. Position-dependent letter-probability matrices of each TF binding motif were kindly provided by Minch and colleagues (*Minch et al., 2015*), who derived them from a ChIP-Seq experiment on virulent *M. tuberculosis* type strain H37Rv. We then ran FIMO using each TF motif on the context FASTA file with a threshold p-value of 0.01. For comparison we also scanned for TF motifs in the context sequences of consistently methylated loci (consistently methylated loci here defined as loci present in at least 30 isolates and methylated in at least 95% of those isolates). Custom scripts then parsed the FIMO output files for each TF binding motif and counted the number of methylated loci and the number of hypomethylated loci matching each TF with a q-value of at least 0.1.

To genotype the transcription factor MntR in each isolate, we used the same method used for the MTase genotyping. The wild-type sequence of *mntR* (Rv2788) was taken from the annotated reference genome H37Rv. BLASTn aligned the wild-type sequence against all genes predicted in each isolate by Prodigal (*Hyatt et al., 2010*). Each matching nucleotide sequence was translated into an amino acid sequence using transeq (EMBOSS 6.6.0.0, available online at www.ebi.ac.uk/Tools/emboss/transeq/index.html) to obtain nonsynonymous variants and truncations. The amino acid sequences were then aligned using MAFFT (*Katoh and Standley, 2013*) v7.205 with the –clustalout option, and a custom script converted the alignment to a genotype.

## Proximal motif site search

Multiple MTase motif sites in close proximity targeted by the same MTase can also potentially cause hypomethylation (*Casadesús and Low, 2013*). To find potential cases of this phenomenon, for each isolate a custom R script read a CSV file with the genome positions of each MTase motif site in the isolate. For each MTase motif site, the script found the nearest MTase motif site of the same type (MamA, MamB, and HsdM) and the same strand, then recorded the nucleotide distance between them. If the distance was less than 100 bp, the sites were considered neighbors ('Nearby Motif,' *Table 1*).

## RNA-Seq analysis

Supplementary table 9 of https://www.nature.com/articles/s41467-019-11948-6 was downloaded and merged by locus tag with our annotated promoters for HsdM. A Benjamini-Hochberg adjusted p-value threshold of 0.05 was set as the criteria for being considered 'differentially expressed', using the column labeled 'padj (BH)'. Two-sided Fisher's exact test was implemented in R to test for independence of HsdM motif site presence in the promoter and differentially expressed genes following

*hsdM* knockout. Genes with an HsdM-targeted adenine within 50 bp upstream of the TSS were considered to have an HsdM promoter motif.

## Acknowledgements

We acknowledge Mr. James O'Neill and Mr. Norman Kuo for their help in curating characterized sigma factor binding sites, and Mr. Afif Elghraoui for instructive discussions and feedback on the manuscript. We thank Drs. Sara Byfors, Ramona Groenheit, Jim Werngren, and Mikael Mansjö from the Public Health Agency of Sweden in Stockholm, Drs. Antonino Catanzaro and Timothy Rodwell from the University of California in San Diego, and Dr. Leen Rigouts from the Institute for Tropical Medicine in Antwerp, Belgium for sharing clinical *M. tuberculosis* isolates with LPCDRP. We acknowledge Dr. Jonas Korlach of Pacific Biosciences for constructive feedback for interpreting results.

## Additional information

### Funding

| Funder | Grant reference number | Author |
|---|---|---|
| National Institute of Allergy and Infectious Diseases | R01AI105185 | Samuel J Modlin<br>Derek Conkle-Gutierrez<br>Calvin Kim<br>Scott N Mitchell<br>Christopher Morrissey<br>Sarah M Ramirez-Busby<br>Sven E Hoffner<br>Faramarz Valafar |

The funders had no role in study design, data collection and interpretation, or the decision to submit the work for publication.

### Author contributions

Samuel J Modlin, Conceptualization, Data curation, Software, Formal analysis, Supervision, Validation, Visualization, Methodology, Writing - original draft, Writing - review and editing; Derek Conkle-Gutierrez, Conceptualization, Software, Formal analysis, Validation, Visualization, Methodology, Writing - original draft, Writing - review and editing; Calvin Kim, Software, Formal analysis, Visualization, Writing - original draft, Writing - review and editing; Scott N Mitchell, Data curation, Software, Formal analysis, Visualization, Writing - original draft, Writing - review and editing; Christopher Morrissey, Software, Formal analysis, Validation, Writing - review and editing; Brian C Weinrick, William R Jacobs, Writing - review and editing; Sarah M Ramirez-Busby, Writing - original draft, Writing - review and editing; Sven E Hoffner, Resources, Supervision, Writing - review and editing; Faramarz Valafar, Conceptualization, Resources, Supervision, Funding acquisition, Investigation, Methodology, Project administration, Writing - review and editing

### Author ORCIDs

William R Jacobs https://orcid.org/0000-0003-3321-3080
Sarah M Ramirez-Busby http://orcid.org/0000-0001-7455-1903
Faramarz Valafar https://orcid.org/0000-0002-3648-9384

### Decision letter and Author response

Decision letter https://doi.org/10.7554/eLife.58542.sa1
Author response https://doi.org/10.7554/eLife.58542.sa2

## Additional files

### Supplementary files

• Supplementary file 1. Activity of observed methyltransferase genotypes. For each distinct methyltransferase (MTase) variant found in our MTBC isolates, we measured the resulting sequencing

kinetics signals of bases targeted by the MTase motif in that isolate, and from them inferred the activity of the variant MTase, reported here. Variants that were not present in our dataset could potentially be with respect to H37Rv instead of a wild-type MTase. *R47W and G154D were only found in H37Rv and H37Ra. **Inferred to be deleterious, since only found in conjunction with D59G and V616A, which result in wild-type methylation patterns in the absence of this insertion. ***Also inferred to be deleterious, since only found in conjunction with V616A. ****K458N only found in tandem with E481A. *****Consensus activity should be taken as tentative, since this genotype was not observed in our study, and some previously reported as loss-of-function were revealed to be partially active with our method of examining all motif instances. 'unknown' means that the effect of the specified mutation cannot be inferred because it does not occur in isolation. †For *hsdM* variant E481A, our study sequenced the same isolate as Chiner-Oms and colleagues, but our genotyping showed both E481A and K458N in *hsdM*, while they only reported E481A. Both studies showed a mild reduction in HsdM activity for this isolate, but it is unclear which mutation causes the reduction, or whether the effect is epistatic.

• Supplementary file 2. MTBC isolates by MTase genotype and activity. Methylation activity for the three MTBC m6A DNA methyltransferases and the genotypes of the genes encoding the proteins that comprise them. Each row corresponds to an isolate.

• Supplementary file 3. Shared MTase motif loci Common MTase motif loci present in > 74 isolates. MTase motif site loci were assigned by our methylome annotation pipeline, using characterized MTase target motifs, and nearby genes transferred from H37Rv references by Rapid Annotation Transfer Tool (RATT).

• Supplementary file 4. Methylation anomalies Microsoft Excel file containing hypervariable and differentially variable motif sites present in > 74 isolates. Sheet one contains all hypervariable motif sites, defined as any site more than 3 SDs above the mean SD size across isolates, calculated using the distribution of $\log_2$ (IPD Ratios) at mamB motif sites. It specifies the coordinates with respect to proximal TSSs and CDS, as well as the standard deviation and median across motif sites of the specified genotype. Sheets 2 and 3 hold the CDSs and TSSs implicated in these analyses and the particular set (s) of variable motif sites they belong to.

• Supplementary file 5. Hypomethylation analysis Consistently hypomethylated MTase motif site loci across 93 clinical isolates. MTase motif site loci were assigned by our methylome annotation pipeline, using proximal H37Rv gene references transferred by Rapid Annotation Transfer Tool (RATT). Consistently hypomethylated loci were classified as unmodified by our Bayesian analysis in a significant number of isolates in which the relevant MTase was mostly active. Significance was calculated using cumulative binomial test, setting the number of MTase-active isolates where a locus was present as the number of trials, and the number of said isolates where the locus was hypomethylated as the number of successes. At a 0.01 significance level, the threshold p-value for significance was 4.72E-07, after a Bonferroni correction for the number of loci tested. Sheet one contains hypomethylated MamA motif site loci. Sheet two contains the hypomethylated MamB loci, and sheet three contains hypomethylated HsdM loci.

• Supplementary file 6. Promoter methylation patterns Frequencies of MTase motif sites relative locations upstream of TSSs in putative promoters and motif sites present across most isolates. Sheets 1 and 3 are organized locus-wise, with frequencies of common loci and their overlap with Sigma-factor consensus motifs are each locus for –10 and –35 consensus sigma factor binding sites. Sheets 2 and 4 use the same underlying data as 1 and 3 but report the frequencies of relative distances upstream of TSS that MTase motifs and SFBSs overlap. Sheet five is similar to sheets 1 and 3, but instead reports all MTase motif sites within 100 bp of TSSs, rather than only the subset that overlap SFBSs. Sheet six lists all loci found across 30 or more isolates that fall within 50 base pairs upstream of a TSS, and for each Sigma Factor lists whether it overlaps with the –10 or –35 motifs, and whether it overlaps with a sigma factor in an arrangement indicative of a true promoter. Sheet seven is similar to six but each row represents a promoter containing at least one MTase motif.

• Supplementary file 7. HsdM knockout RNAseq re-analysis.

• Supplementary file 8. Fasta file containing the nucleotide sequence for insertion sequence IS6110.

• Transparent reporting form

## Data availability

Sequencing data for all M. tuberculosis clinical strains analyzed in this study are deposited under Bio-Project accessions PRJNA555636 and PRJEB8783. All data generated or analyzed for this study are included in the manuscript and supporting files. Source data files have been provided for figures and tables.

The following previously published datasets were used:

| Author(s) | Year | Dataset title | Dataset URL | Database and Identifier |
|---|---|---|---|---|
| Minch K | 2015 | The DNA-binding network of Mycobacterium tuberculosis ChIP-seq dataset | https://www.ncbi.nlm.nih.gov/bioproject/?term=PRJNA255984 | NCBI BioProject, PRJNA255984 |
| Elghraoui A, Modlin SJ, Valafar F | 2017 | Mycobacterium tuberculosis H37Ra genome sequencing and assembly | https://www.ncbi.nlm.nih.gov/bioproject/329548 | NCBI BioProject, SRX1959957, SRX1959958 |
| Berney M | 2015 | Determining the methylome of Mycobacterium tuberculosis | https://www.ncbi.nlm.nih.gov/bioproject/?term=PRJEB8783 | NCBI BioProject, PRJEB8783 |
| Valafar F | 2020 | Mycobacterium tuberculosis reference-quality clinical genomes | https://www.ncbi.nlm.nih.gov/bioproject/?term=PRJNA555636 | NCBI BioProject, PRJNA555636 |

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
