## [Decision Letter]

**Acceptance summary:**

This study is interesting and important because by using PacBio sequencing to conduct SMRT-seq based methylome analyses of 93 *M. tuberculosis* isolates, the authors were able to uniquely provide a detailed description of DNA methylations of *M. tuberculosis* genomes associated with the presence or absence of certain MTases.

**Decision letter after peer review:**

Thank you for submitting your article "Epigenetic mosaicism in the *Mycobacterium tuberculosis* methylome enables phenotypic plasticity without genetic mutation" for consideration by *eLife*. Your article has been reviewed by Dominique Soldati-Favre as the Senior Editor, a Reviewing Editor, and three reviewers. The following individuals involved in review of your submission have agreed to reveal their identity: Josep Casadesús (Reviewer #1).

The reviewers have discussed the reviews with one another and the Reviewing Editor has drafted this decision to help you prepare a revised submission.

Summary:

In this manuscript, the authors used PacBio sequencing to conduct SMRT-seq based methylome analyses of 93 *M. tuberculosis* isolates. This study is interesting and important because it provides a detailed description of DNA methylations of *M. tuberculosis* genomes associated with the presence or absence of certain MTases.

However, the reviewers agreed that there were a number of concerns that need to be addressed. In particular, (1) the authors' argument relating the roles of the methylation patterns in physiology and adaptation are too strong and not supported by the data in this manuscript, (2) the details of the methylation analysis were not clearly described in enough detail, and (3) the "mosaic" finding itself is not new or novel, which needs to be acknowledged and addressed. The storyline of the paper needs to be substantially changed to be more descriptive and data-based (instead of speculation-based). The manuscript should also be edited to be clearer and more concise for the reader as well.

Essential revisions:

1) Add detailed method descriptions for the analysis portions of the manuscript.

2) The physiological consequences of mycobacterial gene expression control by DNA methylation remain to be identified. Therefore, the benefits of phenotypic heterogeneity remain hypothetical. This caveat should be emphasized throughout the manuscript. As support, the authors might perhaps cite game theory papers indicating that phenotypic heterogeneity can be an adaptive strategy in bacterial populations. Here are a few examples (among others):

Wolf, Vazirani and Arkin, 2005.

Ficici and Pollack, 2007.

Lambert and Kussell, 2014.

3) Results subsection “Diverse mutations drive DNA methyltransferase activity profiles”. The interpretation that intermediate IPD ratios are due to heterogeneous methylation looks reasonable. However, heterogeneous methylation is naturally found in batch cultures during DNA replication unless the bacterial culture is synchronized, which obviously is not the case here. Because formation of hemimethylated sites occurs at different regions in individual bacteria, the DNA hemimethylation patterns vary from cell to cell. This can introduce noise into SMRT-sequencing, and in this study it might lead to overvaluation of cell-to-cell DNA methylation heterogeneity. The risk of hemimethylation-associated noise decreases in non-dividing cells. The growth stage of the culture used for SMRT-sequencing is not indicated in the manuscript (the authors cite Elghraoui, Modlin and Valafar, 2017, which does not give details). If non-dividing cells were used, it should be clearly indicated. Otherwise, evidence for stochastic DNA methylation would be less compelling.

4) Results subsection “Transcription factor occlusion explains most hypomethylated sites” and thereafter. The statement that transcription factor occlusion explains most hypomethylated sites should be toned down. The data presented are based on bioinformatic analysis only. Proof of transcription factor binding requires biochemical evidence (e. g., gel shift analysis or DNA footprinting). Without proof of this kind, predictions on DNA binding by transcription factors are speculative.

5) Discussion section. Gries et al. (2010) and Saecker et al. (2002) are wrong citations, not appropriate to support the statement that DNA methylation alters biophysical properties that tune promoter strength. These papers do not mention DNA methylation. Papers that address the effects of DNA adenine methylation on DNA structure include Diekmann (1987), Polaczek et al. (1998) and Kimura et al. (1989).

6) To improve the manuscript to better convey a clear message, reorganize and revise to highlight the focus on MTase mutants and their effects on methylation:

a) MTases' genotyping from WGS and their activities from IPD ratio-based analysis; as well as their demonstration in population, i.e. phylogeny tree;

b) MTases-based methylation heterogeneity from native IPD value-based analysis, suggesting to move Figure 3E, 3F, and 3G (H37Rv-metA, less important) to supplemental and to move Figure 3—figure supplement 1 (MamB) back to Figure 3 with mention of HsdM in the text;

c) MTases-based comparative methylomics for hypervariable sites and hypomethylated sites, and their potential association with transcription factors;

d) MTases-based comparative methylomics for promoters and SFBSs, including RNAseq re-analysis.

7) The use of knock-down (KD) and knock-out (KO) is usually for change of gene/protein quantity or expression level. It may be more appropriate for the authors to use high- (WT), low- (KD), and in- (KO) active mutants for mutation-caused changes of MTase activity.

8) Epigenetic mosaicism is somewhat associated with MTase activity, high or low, and MTase activity is certainly linked to genetic mutation or genotype. As such, "without genetic mutation" in the title and throughout the manuscript is not accurate, and should be toned down.

9) Results subsection “Virulent *M. tuberculosis* type strain H37Rv poorly represents methylomes of recent clinical isolates”: while H37Rv was recognized as a poor reference for methylome, a better one from the assembled/analyzed should have been suggested/used. Please specify the reference if used.

10) Since only SNP analysis was considered for phylogeny tree, the effect of insertion/deletion to methylome and methylomics analysis should be mentioned in Discussion section.

[Editors' note: further revisions were suggested prior to acceptance, as described below.]

Thank you for resubmitting your work entitled "Drivers and sites of diversity in the DNA adenine methylomes of 93 *Mycobacterium tuberculosis* complex clinical isolates" for further consideration by *eLife*. Your revised article has been evaluated by Dominique Soldati-Favre (Senior Editor), a Reviewing Editor and two peer reviewers.

The manuscript has been improved but there are some remaining issues that need to be addressed before acceptance, as outlined below:

The reviewers all agree that the revisions have strengthened the original story and the authors' responses to the reviewers look reasonable. Most importantly, problematic data and/or statements have been either clarified or removed in the revised manuscript. There are a few text revisions that the authors should address before formally accepting the manuscript, as detailed here:

1) The name HsdM should not be used for an orphan DNA methyltransferase. The acronym hsdM has been used for decades to designate the DNA methyltransferase subunit of type I restriction-modification systems (see, for instance, Loenen et al., 2014). I perfectly understand that the authors use HsdM in accordance with previous literature, and I am aware that a name change can cause complications and confusion. Despite this inconvenience, the enzyme should be renamed to make it clear that it is not a subunit of a R-M system. How about renaming the enzyme at or near the end of the manuscript and citing Loenen et al., or any another review on restriction enzymes?

2) Introduction. The authors write that recent SMRT-sequencing studies have revealed that DNA methylation has roles beyond restriction-modification. This statement is unfair to the literature. In γ-proteobacteria, Dam methylation has been known to control DNA replication, mismatch repair, transposon activity and regulation of transcription since the 1980's and roles in bacterial pathogenesis were described in the late 1990s. In α-proteobacteria, control of the cell cycle by CcrM methylation was described in the 1990s and roles in pathogenesis in the first years of the century. Please modify the sentence. SMRT has been a fantastic breakthrough but not the beginning of the story!

3) Results section. Please change "inactive mutation" to "loss-of-function mutation". "Inactive" may suggest that the mutation does not do anything while the actual fact is the opposite.

4) Results section. When you mention that hypomethylated sites have been detected in bacteria different from *M. tuberculosis*, you cite Blow, 2016 (which is fine) and Minch et al., 2015, which does not have anything to do with the subject (DNA methylation is not even mentioned in the paper!). Again, it might be advisable to be fair to the literature citing at least one of the pioneer studies that detected DNA hypomethylation in a bacterial genome. For instance, Ringquist and Smith, 1992 and/or Wang and Church, 1992. Alternatively, you might cite a review.

---

## [Author Response]

Summary:In this manuscript, the authors used PacBio sequencing to conduct SMRT-seq based methylome analyses of 93 M. tuberculosis isolates. This study is interesting and important because it provides a detailed description of DNA methylations of M. tuberculosis genomes associated with the presence or absence of certain MTases.However, the reviewers agreed that there were a number of concerns that need to be addressed.

We thank the reviewers for their insight, attention to detail, and well-articulated feedback. Edits to address revisions are addressed point-by-point below, but to address these overarching concerns, we have made the following changes in revising our manuscript:

In particular, (1) the authors' argument relating the roles of the methylation patterns in physiology and adaptation are too strong and not supported by the data in this manuscript,

We accept this sentiment, and have made revisions accordingly, both in the wording of key sentences and in the manuscript’s scope. We acknowledge that our discussion about the roles methylation patterns play in physiology and adaptation are primarily speculation and hypothesis-generating. We have replaced much of this speculative discussion and hypothesis-generating analysis with more descriptive text and cut speculative discussion to reduce its share of the scope of the manuscript. Relevant revisions are enumerated in our reply to Essential revision #2.

However, we do present data regarding the arrangement of methylation with respect to established regulatory elements. While this arrangement does not confirm any physiological effect of these methylation patterns, it does suggest hypothesis for further experimentation. Thus, while we have cut the speculative portions significantly, we have retained an abridged presentation (Discussion section) of the physiological processes and adaptive mechanisms that our integration of DNA adenine methylomes with annotated complete assemblies support as potentialities previously undescribed in the literature. Throughout the revised manuscript, we have been sure to refer to the data supporting the underlying premise of hypotheses/speculation, and to make clear distinction between hypotheses, speculation, and contentions backed by data presented in this manuscript.

(2) the details of the methylation analysis were not clearly described in enough detail,

We have added substantially to the methods describing how we carried out the methylation analysis. Specific changes are enumerated in our response to Essential revision #1.

and (3) the "mosaic" finding itself is not new or novel, which needs to be acknowledged and addressed. The storyline of the paper needs to be substantially changed to be more descriptive and data-based (instead of speculation-based). The manuscript should also be edited to be clearer and more concise for the reader as well.

We appreciate this perspective and input, as we had wrestled with how to define the scope of this manuscript during its development as well. Upon reflection, we agree that the revised manuscript should be more descriptive, and less speculation-based. We have made changes throughout the manuscript in this spirit, which is captured by our revision of the manuscript title: “Drivers and sites of diversity in the DNA adenine methylomes of 93 *Mycobacterium tuberculosis* complex clinical isolates”. With this title, we reduce focus on the finding of Intercellular Mosaic Methylation and effects of phenotype or adaptivity, broadening the subject of the title to the other findings backed by data presented in the manuscript. Namely, these are:

1) The factors demonstrated (MTase genotype) and implicated (transcription factors) by our data and analysis in driving heterogeneity in DNA adenine methylomes within and across isolates.

2) The sites in the genome where DNA adenine methylation varies across isolates and/or within isolates (the hypomethylated and hypervariable motif sites).

In accordance with the reviewers’ request for a focus more restricted to findings from the data presented in the manuscript, the title change also now only refers to the phenomenon as observed across the isolates analyzed in this study, rather than generalizing to the species level. Our storyline and manuscript organization have also changed accordingly (see replies to Essential revision #6 for specific changes).

However, we believe that while intracellular stochastic methylation has been reported previously, its manifestation across cells has not been formally described. And while some species have been characterized with multiple phase-variable MTases that, through independent variation, create numerous potential combinations of MTases and resulting methylation patterns, we believe that the diversity of methylation patterns conferred by IMM as we describe in the manuscript remains distinct. The most important distinction is that IMM creates a diversity of methylation patterns that scales logarithmically with the number of methylation *sites* targeting by the mosaicking MTase, whereas the previously described systems of multiple phase-variable MTases scale logarithmically with the *number of phase-variable MTases.* The number of sites is at least an order of magnitude greater than the number of phase-variable MTases, and thus creates far more potential methylome states.

That said, we wholeheartedly agree that a mosaic nature has been described in the DNA adenine methylomes of other prokaryotes, thus requiring acknowledgment and discussion in our manuscript. We have added such a discussion (Discussion section) of mosaic-like patterning of other prokaryotic methylomes. We have also included a figure (Figure 9, Discussion section) that clearly lays out where our characterization of Intercellular Mosaic methylation implicated by our data departs from these previous descriptions.

Essential revisions:1) Add detailed method descriptions for the analysis portions of the manuscript.

We thank the reviewers for pointing out that the methods underlying some portions of the analysis could stand to be more explicitly described. We have updated the methods for the following sections and line numbers:

Subsection “Lineage Determination” now includes a brief description of MiruHero’s input and function.

Subsection “Genome annotation” now includes a brief description of the Rapid Annotation Transfer Tool’s inputs, output, and function. The section also now has a citation for our annotation of reference genome H37Rv.

Subsection “Phylogeny construction and mapping of MTase genotypes,” now titled “Phylogenetic analysis,” now clarifies how the MTase genotypes of each isolate were input into iTol for visualization of our phylogenetic trees.

Subsection “Analysis of sequencing kinetics” and subsection “Finding methylation anomalies” have been expanded into the sections “Determination of interpulse duration of MTase motif sites”, “Characterization of MTase activity of MTase Genotypes”, and “Identification of hypervariable MTase motif loci” with additional details.

Subsection “Methylome Annotation,” now named “MTase motif locus assignment,” now contains a detailed description of the information contained in the assigned MTase locus tags, and more detail on how the tags were assigned.

Subsection “Characterizing the kinetic error profile across technical replicates” now specifies the statistical test used.

Subsection “Identification of promoters” now contains more detail on the files and criteria used to identify probable gene promoters.

Subsection “Proximal motif site search” now includes the threshold nucleotide distance and other criteria used to classify which MTase motif sites were neighbors.

Subsection “RNA-Seq analysis” now clarifies how the datasets were merged.

2) The physiological consequences of mycobacterial gene expression control by DNA methylation remain to be identified. Therefore, the benefits of phenotypic heterogeneity remain hypothetical. This caveat should be emphasized throughout the manuscript. As support, the authors might perhaps cite game theory papers indicating that phenotypic heterogeneity can be an adaptive strategy in bacterial populations. Here are a few examples (among others):Wolf, Vazirani and Arkin, 2005.Ficici and Pollack, 2007.Lambert and Kussell, 2014.

We agree with the reviewers that it is important this caveat be made clear. We have now added a paragraph to the Discussion section stating this caveat clearly and discussing the consequences. We have also changed much of the speculative content to descriptive content, and now frame the potential regulatory interactions we have identified as a resource for hypothesis generation, replacing formerly speculative text. We have cited one of the papers you alerted us to in support of the idea that phenotypic heterogeneity may confer survival benefit.

3) Results subsection “Diverse mutations drive DNA methyltransferase activity profiles”. The interpretation that intermediate IPD ratios are due to heterogeneous methylation looks reasonable. However, heterogeneous methylation is naturally found in batch cultures during DNA replication unless the bacterial culture is synchronized, which obviously is not the case here. Because formation of hemimethylated sites occurs at different regions in individual bacteria, the DNA hemimethylation patterns vary from cell to cell. This can introduce noise into SMRT-sequencing, and in this study it might lead to overvaluation of cell-to-cell DNA methylation heterogeneity. The risk of hemimethylation-associated noise decreases in non-dividing cells. The growth stage of the culture used for SMRT-sequencing is not indicated in the manuscript (the authors cite Elghraoui, Modlin and Valafar, 2017, which does not give details). If non-dividing cells were used, it should be clearly indicated. Otherwise, evidence for stochastic DNA methylation would be less compelling.

We thank the reviewers for pointing out this alternative cause of heterogeneous methylation. Our Results section now considers transient hemimethylation as a potential cause of heterogeneous methylation, in addition to phase variant MTase activity and intracellular stochastic methylation. Our isolates were sequenced after reaching Stationary Phase, which our subsection “Sample preparation and extraction” now indicates. While it is still likely that a subpopulation of cells were undergoing or had recently undergone cell division during DNA extraction, we believe the results of our read-level analysis with SMALR rules out transient hemimethylation as the primary cause of the observed heterogeneity. SMALR’s SMp method calculates a native IPD value for each sequencing read in an isolate, by averaging the IPD values across multiple target sites in the read matching a specified motif. This allows a distinction between fully methylated reads, fully unmethylated reads, and reads with a mix of methylated and unmethylated sites. Hemimethylation would likely result in a mix of fully methylated and fully unmethylated reads (or a mix of fully methylated and partially methylated reads, if the MTase performed distributive rather than processive methylation). The authors of SMALR have indeed observed that a small fraction (0.07%) of sequencing reads were fully unmethylated in cultures of bacterial strains with fully active MTase alleles, and attributed this to transient hemimethylation. The heterogeneity we observed in *mamA*_E270A_ and *mamA*_G152S_ variant isolates was not consistent with this transient hemimethylation. Their distribution of native IPD values were consistent with the majority of reads being partially methylated (with only a fraction of sites methylated in each read), rather than a mix of fully methylated and unmethylated reads. Additionally, if the heterogeneous methylation was only the result of hemimethylation during culturing, it would not be exclusive to specific MTase genotypes.

4) Results subsection “Transcription factor occlusion explains most hypomethylated sites” and thereafter. The statement that transcription factor occlusion explains most hypomethylated sites should be toned down. The data presented are based on bioinformatic analysis only. Proof of transcription factor binding requires biochemical evidence (e.g., gel shift analysis or DNA footprinting). Without proof of this kind, predictions on DNA binding by transcription factors are speculative.

The subsection “Transcription factor occlusion explains most hypomethylated sites” is now named “Sequence context of most hypomethylated sites are consistent with transcription factor occlusion.” The section now makes it clear that transcription factor binding to these hypomethylated MTase motif sites has not been confirmed.

5) Discussion section. Gries et al. (2010) and Saecker et al. (2002) are wrong citations, not appropriate to support the statement that DNA methylation alters biophysical properties that tune promoter strength. These papers do not mention DNA methylation. Papers that address the effects of DNA adenine methylation on DNA structure include Diekmann (1987), Polaczek et al. (1998) and Kimura et al. (1989).

We thank the reviewers for catching this oversight and for providing helpful references pertinent to the unsupported elements of our sentence. We have added a more appropriate reference.

6) To improve the manuscript to better convey a clear message, reorganize and revise to highlight the focus on MTase mutants and their effects on methylation:a) MTases' genotyping from WGS and their activities from IPD ratio-based analysis; as well as their demonstration in population, i.e. phylogeny tree;b) MTases-based methylation heterogeneity from native IPD value-based analysis, suggesting to move Figure 3E, 3F, and 3G (H37Rv-metA, less important) to supplemental and to move Figure 3—figure supplement 1 (MamB) back to Figure 3 with mention of HsdM in the text;c) MTases-based comparative methylomics for hypervariable sites and hypomethylated sites, and their potential association with transcription factors;d) MTases-based comparative methylomics for promoters and SFBSs, including RNAseq re-analysis.

We thank the reviewers for suggesting this renewed organization to clarify the message of our manuscript. We have partially adopted the suggested organization, and pruned some sections that did not fall into any of A-D. We agree that a focus on MTase mutants’ brings clarity to A and B and have adjusted the text to more clearly reflect that. C and D, however, identify sites of methylomic variation that are either independent of genotype (organization of MTase motifs with respect to promoters) or are stratified by the level of MTase activity (which can include multiple MTase genotypes), rather than MTase genotype (identification of hypomethylated and hypervariable sites). Therefore, we have not organized C and D according to MTase genotype.

7) The use of knock-down (KD) and knock-out (KO) is usually for change of gene/protein quantity or expression level. It may be more appropriate for the authors to use high- (WT), low- (KD), and in- (KO) active mutants for mutation-caused changes of MTase activity.

All instances of “knocked down” have been replaced with “partially active.” All instances of “knockout” have been replaced with “inactive,” except when referring to the previous mutagenesis experiments on *hsdM* and m*etA*. Some uses of “wild-type” were replaced with “fully active.” Other uses of “wild-type” were kept to distinguish from variants that do not cause a reduction in MTase activity (such as *mamA*:A34V).

8) Epigenetic mosaicism is somewhat associated with MTase activity, high or low, and MTase activity is certainly linked to genetic mutation or genotype. As such, "without genetic mutation" in the title and throughout the manuscript is not accurate, and should be toned down.

The title has been changed to remove the confusing phrase “without genetic mutation.” Other uses of the phrase have also been removed throughout the manuscript.

9) Results subsection “Virulent M. tuberculosis type strain H37Rv poorly represents methylomes of recent clinical isolates”: while H37Rv was recognized as a poor reference for methylome, a better one from the assembled/analyzed should have been suggested/used. Please specify the reference if used.

We thank the reviewers for this consideration. Upon reflection, we concluded that it was confusing to include this section and have removed it. Our analysis did not rely on a reference methylome, instead comparing methylation levels across isolates at common homologous sites. The identification of common homologous sites for cross-isolate comparisons is now described in more detail in the subsection “MTase motif locus assignment.”

10) Since only SNP analysis was considered for phylogeny tree, the effect of insertion/deletion to methylome and methylomics analysis should be mentioned in Discussion section.

We thank the reviewers for alerting us to the lack of clarity regarding the methods for MTase genotyping and how our analyses used the phylogenetic tree. Our MTase genotyping did not use the phylogenetic tree, and was able to detect indels, such as the insertion of IS6110 into *mamB*. Our genotyping methods are described in methods under “MTase Genotyping.” The subsection “Phylogeny construction and mapping of MTase genotypes” was vaguely worded and may have caused confusion. This subsection, now titled “Phylogenetic analysis” now clarifies how the MTase genotypes of each isolate were input into iTol for visualization of our phylogenetic trees.

[Editors' note: further revisions were suggested prior to acceptance, as described below.]

The reviewers all agree that the revisions have strengthened the original story and the authors' responses to the reviewers look reasonable. Most importantly, problematic data and/or statements have been either clarified or removed in the revised manuscript. There are a few text revisions that the authors should address before formally accepting the manuscript, as detailed here:

We thank the reviewers for their feedback and have addressed all considerations below.

1) The name HsdM should not be used for an orphan DNA methyltransferase. The acronym hsdM has been used for decades to designate the DNA methyltransferase subunit of type I restriction-modification systems (see, for instance, Loenen et al., 2014). I perfectly understand that the authors use HsdM in accordance with previous literature, and I am aware that a name change can cause complications and confusion. Despite this inconvenience, the enzyme should be renamed to make it clear that it is not a subunit of a R-M system. How about renaming the enzyme at or near the end of the manuscript and citing Loenen et al., or any another review on restriction enzymes?

We agree with the reviewer’s recommendation to rename *hsdM* and have likewise recommended renaming associated genes *hsdS* and *hsdS.1* to avoid the erroneous implication that they are part of a functional Type 1 Restriction-Modification system. We have added a paragraph near the end of the discussion suggesting the name change and briefly explaining its rationale and referring to literature supporting this change.

2) Introduction. The authors write that recent SMRT-sequencing studies have revealed that DNA methylation has roles beyond restriction-modification. This statement is unfair to the literature. In γ-proteobacteria, Dam methylation has been known to control DNA replication, mismatch repair, transposon activity and regulation of transcription since the 1980's and roles in bacterial pathogenesis were described in the late 1990s. In α-proteobacteria, control of the cell cycle by CcrM methylation was described in the 1990s and roles in pathogenesis in the first years of the century. Please modify the sentence. SMRT has been a fantastic breakthrough but not the beginning of the story!

We agree that the original sentence unfairly represented research on functional roles of prokaryotic DNA methylation discovered well before the advent of SMRT-sequencing. We have revised the sentence to “Prokaryotic DNA methylation has been shown to mediate diverse functions (Hernday et al., 2002; Ringquist and Smith, 1992; Wright et al., 1997), far beyond its originally understood role as the self-protective component of Restriction-Modification systems (RM systems).” Per the reviewer’s suggestions we cite Ringquist and Smith, David Low and Hernday’s 2002 paper on pili switches, and Wright’s 1997 paper on CcrM methylation in α-proteobacteria.

3) Results section. Please change "inactive mutation" to "loss-of-function mutation". "Inactive" may suggest that the mutation does not do anything while the actual fact is the opposite.

We have changed “inactive” to “loss-of-function” here and in a few other places in the manuscript with the same issue (Results section, Materials and methods section).

4) Results section. When you mention that hypomethylated sites have been detected in bacteria different from M. tuberculosis, you cite Blow, 2016 (which is fine) and Minch et al., 2015, which does not have anything to do with the subject (DNA methylation is not even mentioned in the paper!). Again, it might be advisable to be fair to the literature citing at least one of the pioneer studies that detected DNA hypomethylation in a bacterial genome. For instance, Ringquist and Smith, 1992 and/or Wang and Church, 1992. Alternatively, you might cite a review.

We thank the reviewer for catching this erroneous citation. We have now fixed and double-checked the citations throughout the manuscript. For the detection of hypomethylated sites in bacteria we now cite the following papers:

Hernday et al., 2002.

Ringquist and Smith, 1992.

Blow et al., 2016.